

# On the ocean's response to enhanced Greenland runoff in model experiments: relevance of mesoscale dynamics and atmospheric coupling

Torge Martin[1] and Arne Biastoch[1,2]

[1]GEOMAR Helmholtz Centre for Ocean Research Kiel, Kiel, Germany
[2]Christian-Albrechts-Universität Kiel, Kiel, Germany

**Correspondence:** Torge Martin (tomartin@geomar.de)

**Abstract.** Increasing Greenland Ice Sheet–melting is anticipated to impact watermass transformation in the subpolar North Atlantic and ultimately the meridional overturning circulation. Complex ocean and climate models are widely applied to predict magnitude and timing of related impacts under projected future climate. We discuss the role of the ocean mean state, subpolar gyre circulation, mesoscale eddies and atmospheric coupling in shaping the response of the subpolar North Atlantic Ocean to enhanced Greenland runoff. In a suite of eight dedicated 60 to 100-year long model experiments with and without atmospheric coupling, with eddy processes parameterized and explicitly simulated, with regular and significantly enlarged Greenland runoff, we find (1) a major impact by the interactive atmosphere in enabling a compensating temperature feedback, (2) a non-negligible influence by the ocean mean state biased towards greater stability in the coupled simulations, both of which making the Atlantic Merdional Overturning Circulation less susceptible to the freshwater perturbation applied, and (3) a more even spreading of the runoff tracer in the subpolar North Atlantic and enhanced inter-gyre exchange with the subtropics in the strongly eddying simulations. Overall, our experiments demonstrate the important role of mesoscale ocean dynamics and atmosphere feedbacks in projections of the climate system response to enhanced Greenland Ice Sheet–melting and hence underline the necessity to advance scale-aware eddy parameterizations for next-generation climate models.

## 1 Introduction

Watermass transformation in the subpolar North Atlantic (SPNA) plays a key role in the global thermohaline circulation. Warm and salty Atlantic water being transported northward by the Atlantic Merdional Overturning Circulation (AMOC) cools as it loses heat to the atmosphere but also freshens by mixing with polar waters exported from the Arctic Ocean, coastal runoff and precipitation. The AMOC strength is one of the major tipping elements in the climate system (Lenton et al., 2008, 2019; Drijfhout et al., 2015), its poleward heat transport is important to the inhabitability of northern Europe as well as northern hemisphere ice sheet growth and decay. A remnant of larger land ice areas, the present-day Greenland Ice Sheet (GrIS) still



holds a freshwater mass equivalent to about 7.2 m of global sea-level rise, which qualifies the ice sheet as a tipping element itself (Lenton et al., 2008; Aschwanden et al., 2019). Observations and model-supported estimates show an acceleration of GrIS melting over the past couple of decades (Chen et al., 2006; Barletta et al., 2013; Mouginot et al., 2019; The IMBIE Team, 2020) leaving the ice sheet at a negative net mass balance in each of the last 25 years (Polar Portal Season Report 2021, polarportal.dk). This led to an increase in freshwater flux from Greenland of almost 50% from the mid-1990s to 2010 (Bamber et al., 2018). However, Greenland meltwater still makes a relatively small contribution to the SPNA freshwater budget considering Arctic Ocean export (Haine et al., 2015; Steur et al., 2018). Submarine meltwater is hard to trace in Greenland's boundary currents and was not yet detected in deep water formation regions (Rhein et al., 2018; Huhn et al., 2021).

Numerous model studies have demonstrated that enhanced mass loss by the GrIS has the potential to reduce the strength of the AMOC (e.g. Rahmstorf et al., 2005; Swingedouw et al., 2013; Jackson and Wood, 2018; Golledge et al., 2019). The additional freshwater spread across the SPNA stabilizes the water column, eventually limits overall heat loss in the eastern North Atlantic, inhibits deep convection, and thus reduces the amount and density of North Atlantic Deep Water being formed. Uncertainty remains regarding the true sensitivity of the AMOC to global warming and associated additional freshwater input from the GrIS (Rahmstorf et al., 2015; Bakker et al., 2016; Böning et al., 2016; Weijer et al., 2020). Availability of computing power restricts studies of millennial timescales to less comprehensive, non-eddying models (e.g. Rahmstorf et al., 2005; Hawkins et al., 2011) whereas models of full complexity applied to mesoscale-resolving grids are limited to multi-decadal applications (e.g. Böning et al., 2016; Dukhovskoy et al., 2019). Comprehensive climate models indicate robustly that the large-scale ocean circulation responds on a multi-decadal timescale to sudden changes in Greenland meltwater runoff (Weijer et al., 2012; Swingedouw et al., 2013; Jackson and Wood, 2018; Martin et al., 2022). Mesoscale dynamic features, such as eddies and boundary currents, play a critical role in watermass transformation in the SPNA, where for instance eddies advect fresh polar waters from the boundary into the central Labrador Sea and are crucial for the process of restratification after deep convection, and thus need to be simulated or properly parameterized (Gelderloos et al., 2011; Böning et al., 2016; Rieck et al., 2019; Castelao et al., 2019; Georgiou et al., 2019; Tagklis et al., 2020; Pennelly and Myers, 2022). The implications of failing to correctly represent mesoscale processes in the deep convection regions of the SPNA in GrIS melt related studies are not well quantified. In the present study, we aim to bridge the gap between most complex, eddy-rich ocean hindcasts (Böning et al., 2016) limited to the past few decades and comprehensive but non-eddying climate model simulations (Martin et al., 2022) typically applied to idealized freshwater scenarios for centennial projections or millennial paleo applications.

We conduct a freshwater-release experiment of 0.05 Sv enhanced runoff from Greenland using a hierarchy of model configurations. The magnitude of this perturbation is less than in traditional hosing experiments ($\geq$0.1 Sv, e.g. Gerdes et al., 2006; Hu et al., 2011; Weijer et al., 2012; Swingedouw et al., 2013; Jackson and Wood, 2018) and within the range of estimates for the end of the $21^{st}$ century using linear trend extrapolation (0.06–0.08 Sv, Bamber et al., 2018; The IMBIE Team, 2020) or sophisticated ice-sheet modeling of global warming effects under the RCP8.5 scenario (0.018–0.075 Sv Golledge et al., 2019; Goelzer et al., 2020). For comparison, the imbalance in GrIS mass over the passed decade equals an average runoff increase of 0.007–0.008 Sv (Bamber et al., 2018; The IMBIE Team, 2020). We also deliberately separate the effect of the freshwater from any other factors potentially causing an AMOC decline, such as global warming. In this, we neglect the warming that





would have caused the melting in the first place (Mikolajewicz et al., 2007; Golledge et al., 2019). Our approach is tailored to shedding light on the role of atmospheric feedbacks and mesoscale ocean eddies. The baseline simulations are conducted with a comprehensive global climate model with non-eddying ocean on a 1/2° grid. We then carry out the same experiments with
(a) just the ocean component forced by an atmospheric reanalysis product and (b) the horizontal grid refined to 1/10° in the North Atlantic between 30°N and 85°N , which yields a strongly eddying ocean in the region of interest, and (c) both. Such a systematic comparison of coupled vs. forced and eddying vs. non-eddying model configurations was not yet conducted to our knowledge. Similar comparisons using forced ocean simulations of non-eddying to strongly eddying grid resolutions have been performed previously by, for instance, Weijer et al. (2012), Böning et al. (2016) and Dukhovskoy et al. (2016). The effect
of atmospheric coupling was investigated by Stammer et al. (2011) using a non-eddying ocean model component.

Our goal is to demonstrate and attribute the sensitivity of the AMOC to the processes of atmosphere-ocean interaction and mesoscale ocean dynamics. We address the following questions: (1) How important is the mean state for the ocean? (2) Are atmospheric feedbacks a major player in the ocean's response? (3) Does the explicit simulation of mesoscale eddies matter for the response? Our results also provide insight on the suitability of certain model configurations for investigating the ocean
response to enhanced GrIS melting. The hierarchy of model configurations is introduced in the next section along with details of the freshwater perturbation. There, we also provide an overview of the most critical improvements by the grid refinement. Results of the freshwater experiments are presented in Section 3 and discussed in Section 4. A brief summary and concluding remarks are provided in the last section.

## 2 Model configurations and experiment

The Flexible Ocean and Climate Infrastructure (FOCI) at GEOMAR (Matthes et al., 2020) combines the ECHAM6.3 atmosphere (Stevens et al., 2013; Müller et al., 2018) and the JSBACH land models (Reick et al., 2013) with the NEMO3.6 (Madec, 2016) ocean and LIM2 sea-ice models (Fichefet and Maqueda, 1997) using the coupler OASIS3-MCT (Valcke, 2013). Atmosphere (and land) components are applied to a T63 (1.9°) grid with 95 vertical levels reaching up to 0.01 hPa. Ocean (and sea ice) models run on the ORCA05 grid (1/2°) with 46 vertical levels resolving the top 200 m of the water column at 5–
10 m. The coupled FOCI simulations presented here all branch off a 1500-year long pre-industrial control run (internal ID FOCI1.3-SW038) that started with an atmosphere and ocean at rest and was initialized using ocean potential temperature and salinity fields of the PHC3.0 climatology (Steele et al., 2001). Further details of FOCI and the pre-industrial climate control experiments are found in Matthes et al. (2020).

FOCI was specifically designed for applying 2-way high-resolution regional nesting to the ocean component of a coupled
climate model using Adaptive Grid Refinement in Fortran (AGRIF, Debreu et al., 2008). For the VIKING10 nest used here and first introduced in Matthes et al. (2020), the grid refinement is applied to 30–85°N in the Atlantic focusing on subpolar processes while including entire Greenland for the freshwater-release experiments described below (Figure 1a). The nest features a 5-times finer horizontal grid resolution of 1/10°, hence simulates a strongly eddying ocean in large parts of the refinement region. Since the ORCA05 tripolar grid is stretched towards the North Pole, the resolution around Greenland of 24–32 km at





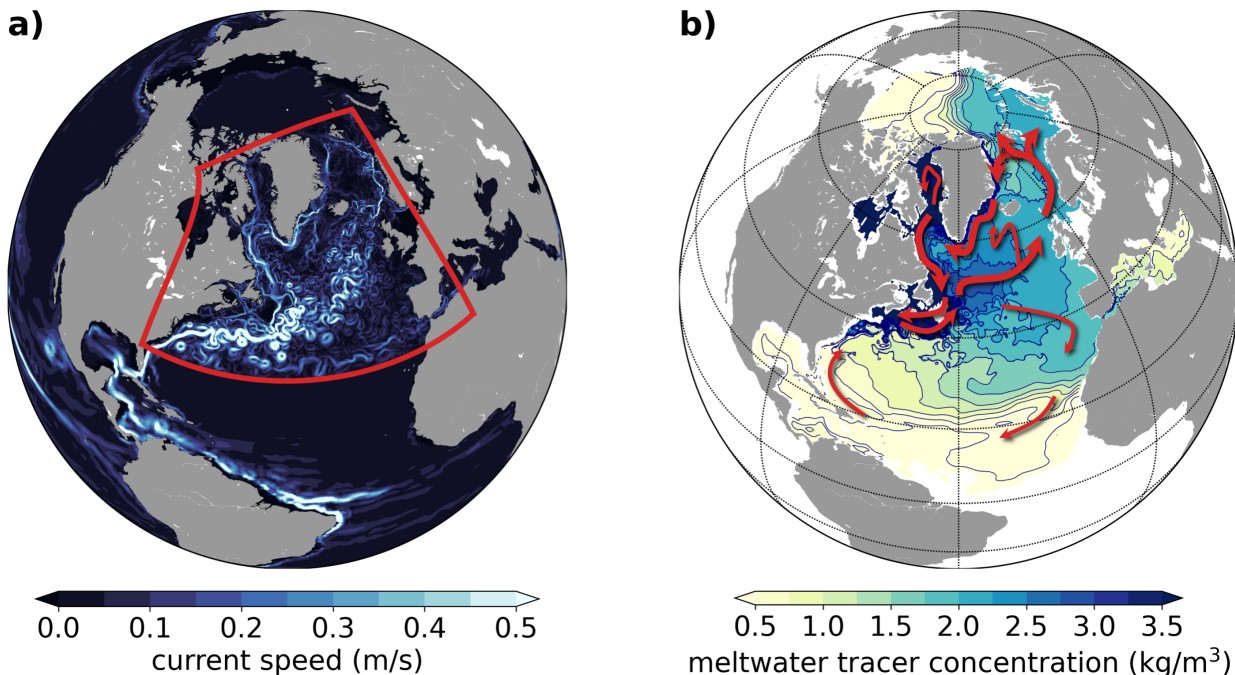

**Figure 1.** (a) Snap shot (5-day mean) of upper ocean current speed at about 100 m depth exemplifying the impact of the grid refinement. The area covered by the 2-way nested model VIKING10 and refined from 1/2° to 1/10° is marked by the red frame. (b) Concentration of the passive tracer tagging the freshwater perturbation (enhanced Greenland runoff) at the end of the experiment after 100 years (same 5-day mean, also at 100 m depth). Red arrows sketch the main pathways of the added Greenland runoff.

1/2° reaches 4.5–6.5 km with the refinement to 1/10° applied. Still, this must be considered only eddy-permitting poleward of about 50° latitude (Smith et al., 2000; Hallberg, 2013). In contrast, with a default 1/2° ocean grid spacing the standard FOCI ocean clearly is non-eddying and the effect of mesoscale eddies is parameterized following Gent and Mcwilliams (1990) applying a space invariant eddy induced velocity coefficient ($rn\_aeiv\_0$) of 1000 m²/s. This also applies to the global 1/2° host model running with the eddying nest. The benefit of resolving mesoscale ocean fronts with the nested configuration for air-sea

fluxes is limited, because the exchange fluxes are computed by the atmosphere model on its coarser grid. Larger scale improvements in surface conditions due to the more realistic ocean dynamics do imprint on the coupled system. For details of the coupling approach with 2-way nesting please see Matthes et al. (2020).

Using the same ocean and sea-ice model, we run ocean-only simulations without and with the VIKING10 nest using CORE-II atmospheric forcing (Large and Yeager, 2009). These hindcast experiments cover the historical period 1948–2009, i.e. extend

over 62 years. The initial state of these simulations, both without and with nest, is generated by initializing the model with the PHC3.0 climatology (Steele et al., 2001) and running a 30-year spinup from 1980 to 2009. Parameter settings are the same





**Table 1.** Overview of numerical experiments: Coupled experiments apply the ECHAM6 atmosphere model, forced ones use CORE-II atmosphere reanalysis. In the $1/2°$ global ocean model eddies are parameterized using GM (Gent and Mcwilliams, 1990) whereas in the North Atlantic nest with a grid resolution of $1/10°$ mesoscale eddies are simulated explicitly instead ($^*$Note, GM is also applied to the global host model of the nested configurations.). A freshwater flux (FWF) of 0.05 Sv is added as spatially and seasonally varying runoff along Greenland's coasts in the perturbation experiments. See main text for details.

| experiments | model configuration | atmosphere representation | ocean grid resolution | eddy representation | climate state | FWF (Sv) | internal run ID |
|---|---|---|---|---|---|---|---|
| reference simulation | coupled | model | $1/2°$ | parameterized | pre-industrial | – | FOCI1.10_TM020 |
| | coupled-nested | model | $1/10°$ | explicit | pre-industrial | – | FOCI1.10_TM026 |
| | forced | reanalysis | $1/2°$ | parameterized | present | – | ORCA05.L46_KTM03p15 |
| | forced-nested | reanalysis | $1/10°$ | explicit$^*$ | present | – | ORCA05.L46_KTM03p25 |
| freshwater perturbation | coupled | model | $1/2°$ | parameterized | pre-industrial | 0.05 | FOCI1.10_TM024 |
| | coupled-nested | model | $1/10°$ | explicit$^*$ | pre-industrial | 0.05 | FOCI1.10_TM028 |
| | forced | reanalysis | $1/2°$ | parameterized | present | 0.05 | ORCA05.L46_KTM03p16 |
| | forced-nested | reanalysis | $1/10°$ | explicit$^*$ | present | 0.05 | ORCA05.L46_KTM03p26 |

as for the coupled model configurations with the exception of applying a weak surface salinity restoring towards the PHC3.0 climatology in the open ocean ($rn\_deds = -33.33$ mm/d, i.e. 180 days for the model's top layer).

In the following, we refer to FOCI simulations as 'coupled', ocean-only experiments as 'forced', and simulations including
the VIKING10 nest as 'nested' (see Table 1 and Fig. 2 for an overview). The latter we also address as 'eddying' or 'strongly eddying' whereas the standard setup without nest is 'non-eddying' or 'eddy-parameterized'. For the analysis and results presented in the following, it is important to understand that model parameters were optimized for the coupled simulations with FOCI and that parameters used for the nested configuration FOCI-VIKING10 are only scaled proportionally according to (nonlinear) dependencies on grid resolution and time stepping. The ocean-only experiments were simply executed with the same set
of parameters as their coupled counterpart—no further "tuning" was undertaken. Coupled and forced simulations differ in their climate state as all FOCI simulations are run under pre-industrial boundary conditions whereas the ocean-only simulations are forced with "historical" atmospheric conditions.

The strongly eddying simulations feature much more pronounced boundary currents and a strongly meandering North Atlantic Current (NAC), see Figure 1a. As a result of the improved dynamics, strength and shape of the subpolar gyre are well
represented and the NAC follows a less zonal, much more realistic path forming the so-called Northwest Corner (Lazier, 1994), a northward and then eastward turning of the currents off Newfoundland and Flemish Cap. This is shown by the standard deviation in sea-surface hight anomalies computed from 5-daily mean model output of a 50-year reference period (Fig. 3a). The realistic simulation of the NAC path nearly eliminates the cold bias, a mismatch of simulated and observed sea-surface temperature (SST) in the mid-latitude North Atlantic, which is prominent feature of the non-eddying simulations (Fig. 3b). Less



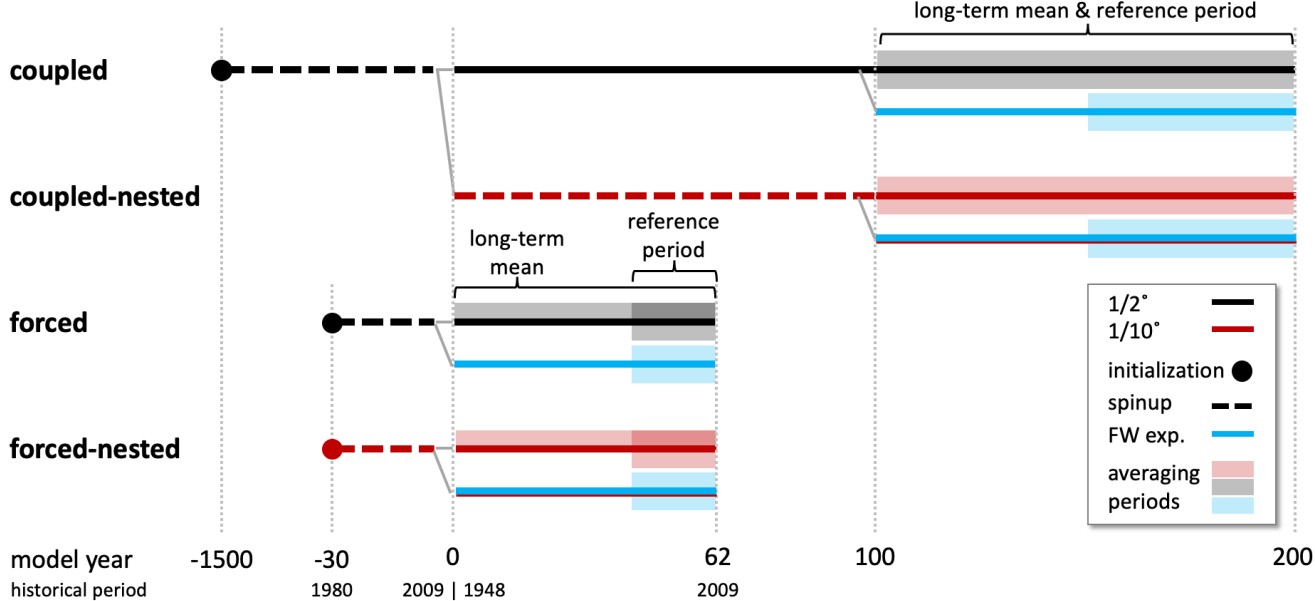

**Figure 2.** Sketch of the experiment setup showing the spinup, branching off and duration of all model simulations. All simulations were initialized using the PHC3.0 ocean climatology (Steele et al., 2001) starting from rest. Coupled non-eddying and strongly eddying experiments share the same low-resolution 1500-year long spinup; coupled freshwater experiments where branched off of the respective control run after 100 years. See main text for details.

often referred to, the cold bias is associated with a fresh bias, a salinity low of up to 2, equally caused by an overestimated south-eastward spreading of polar waters in non-eddying simulations (Fig. 3c). The eddying simulations have saltier overflow water and hence tend towards saltier conditions in the Labrador and Irminger Seas, which promotes intensified, partly overly pronounced deep convection in winter as is discussed in the next section. All of these features concur with earlier studies comparing NEMO-based simulations of similar resolution spread (Treguier et al., 2005; Talandier et al., 2014; Marzocchi et al.,
2015; Hewitt et al., 2020; Koenigk et al., 2021)

   The freshwater-release experiment presented is simplified in both, overall magnitude and its step-like spontaneous onset of the perturbation, but it is most realistic in the spatial distribution and seasonality of the additional freshwater flux inserted along the coast of Greenland. The perturbation is constructed from the monthly-mean runoff plus discharge fluxes of Bamber et al. (2018) by averaging the period 1992–2016, which includes better mass balance estimates and covers the pattern of recently
increased GrIS melting. Spatial heterogeneity and seasonal cycle of the original data are maintained, data extending beyond Greenland is not considered though. The resulting climatology is then scaled to a moderate but significant perturbation of 0.05 Sv on annual mean (see Martin et al., 2022, their Fig. S1). This flux is released as an interannually invariant liquid runoff along Greenland's coast over 62 and 100 in the forced and coupled experiments, respectively, in addition to the simulated (coupled) or prescribed runoff (forced). We note, that this perturbation yields a barystatic sea-level rise of 0.44 m over 100




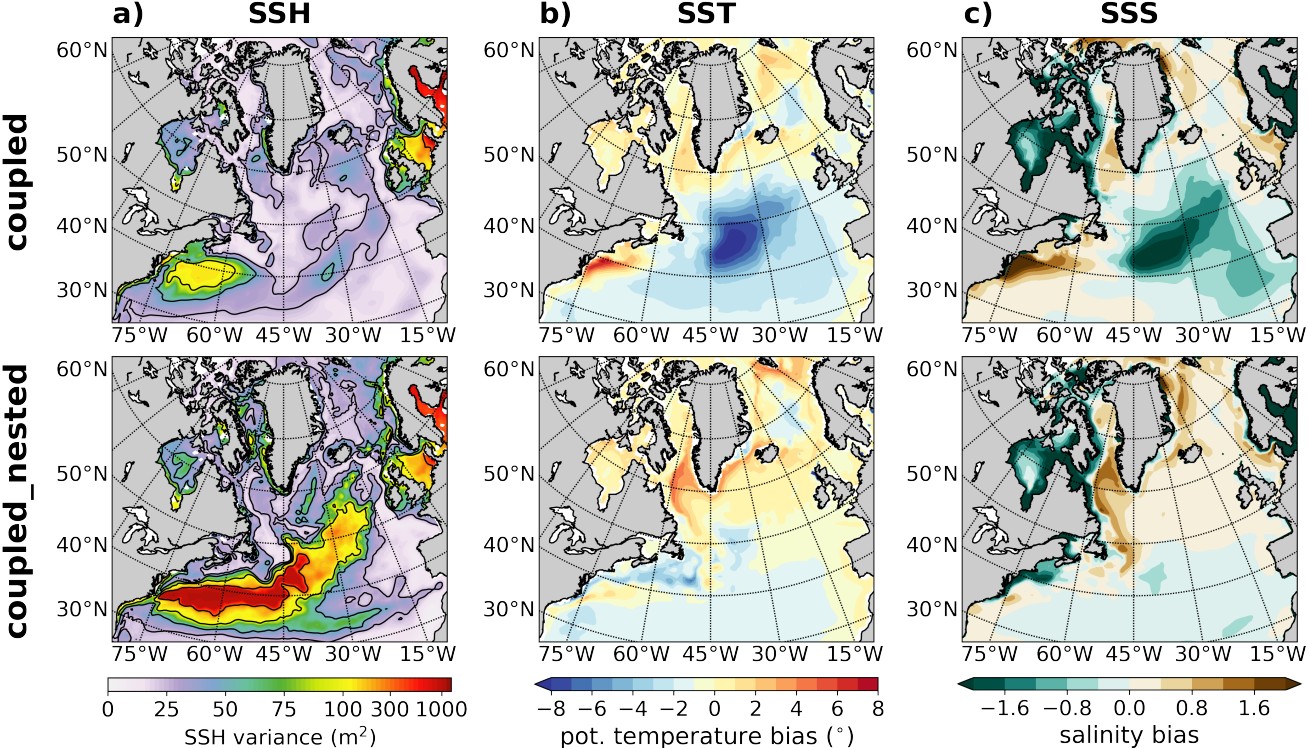

**Figure 3.** Examplary improvements by the regional grid refinement from 1/2° (coupled) to 1/10° (coupled_nested): (a) Enhanced mesoscale activity indicated by sea-surface height (SSH) variance in 5-day mean output, (b) reduced North Atlantic cold bias in sea-surface temperature (SST) and (c) improved representation of sea-surface salinity (SSS). All examples from the coupled model experiments FOCI and FOCI-VIKING10 based on 50-year means. The SST and SSS biases are computed as difference from HadISST (Rayner, 2003), mean over 1870-1899 to compare with pre-industrial model experiments, and WOA98 climatological salinity (Levitus et al., 1998), respectively.

years (Martin et al., 2022), which is about five times the magnitude projected by simulations under the RCP8.5 climate scenario (Goelzer et al., 2020).

We are aware that the way we prescribe the freshwater perturbation is a simplification: applied to the ocean surface layer, maximum runoff in June to August, all liquid runoff. In reality, half of the mass loss of Greenland's glaciers is dynamical discharge (The IMBIE Team, 2020) such as iceberg calving. Further, meltwater runoff is first entrained into the fjord circulation, 140 which causes both, a vertical redistribution and a temporal delay of several weeks before entering the open ocean (Straneo and Cenedese, 2015; Straneo et al., 2016). However, we consider associated errors small for three reasons: Firstly, it is estimated that about half of all icebergs already melt inside the fjords, secondly, Marson et al. (2021) have not found a major impact on deep convection by explicitly simulating icebergs, and thirdly, we find the prescribed freshwater rapidly mixed over the depth of the Greenland shelf by the ocean model also shifting the seasonal peak by a month.



We trace the distribution pathways of the added freshwater by applying a passive tracer. The related model output provides tracer concentration in kg/m$^3$. This means that at any given time, the global tracer mass computed as global sum of the product of tracer concentration and grid-cell volume equals the accumulated runoff from Greenland added as freshwater perturbation. We find an error of <1% and ≤5% after 62 years for the non-nested and nested configurations, respectively, due to the not entirely conservative nature of the NEMO tracer and AGRIF codes for the given setup (e.g. linear free surface). The tracer proves a powerful tool to visualize and understand the pathways of Greenland meltwater redistribution in the ocean as demonstrated in Figure 1b.

## 3 Results

We compute the response to the freshwater perturbation as deviation of the experiment from the reference simulation (perturbation minus control run). As such, the difference is affected by variations in internal variability between the simulations. Variations in internal variability is much more pronounced in the coupled experiments, which evolve freely within the pre-industrial boundary conditions provided, whereas the forced experiments are bound by the historical atmospheric conditions, which are the same for each simulation. Further, mesoscale dynamics simulated explicitly in the strongly eddying experiments enhance ocean internal variability. Lastly, the perturbation experiments all have a certain adjustment period and a new "equilibrium" state can only be expected to exist after several decades. Considering all this, we found optimal signal-to-noise ratios (see Appendix A for details) by using the averaging periods illustrated in Figure 2: In the coupled case we compute the reference mean state over years 101–200 and the perturbed state from the period 151–200. For the forced experiments noise can be optimally reduced by averaging over the same years 43–62 for the reference and perturbed states. To present the control long-term mean state of each model configuration we use years 101–200 and 1–62 for the coupled and forced simulations, respectively. These periods are used throughout this study if not stated otherwise.

We present our results structured by quantities and processes rather than discussing the reference states first and then the responses. This is to highlight the role of the respective mean state on the consequences of the freshwater perturbation.

### 3.1 Ocean mean states and responses

#### AMOC

The strength of the AMOC can be considered a diagnostic integrating all the effects triggered by the additional freshwater input into the SPNA since the overturning is sensitive to changes in temperature, salinity, surface heat flux and deep mixing in this region, all of which will be addressed in the following. We summarize the AMOC mean states and changes in response to the freshwater perturbation of the four configurations in Table 1 and present the associated monthly to interannual variability in Figure 3. AMOC strength is defined here as the maximum of the Atlantic streamfunction in latitude-depth space computed from the zonally averaged meridional velocity of monthly mean model output.

The weakest AMOC is simulated by the non-eddying forced configuration at 13.8 Sv, the coupled-nested setup features the strongest overturning at 21.7 Sv (see Tab. 2). Coupled and forced-nested experiments are relatively close at 16.8 and 17.1 Sv.





**Table 2.** AMOC mean state and response to freshwater perturbation measured by the maximum transport in the overturning stream function computed from zonally averaged Atlantic basin meridional velocities. Mean state and response are computed over years 1–100 (1–62) and 51-100 (43–62) for coupled (forced) experiments, respectively. Additionally, responses for the two coupled experiments computed as average over years 43–62 are provided in brackets. The standard deviation $\sigma$ based on monthly mean model output is provided as a measure of internal variability and noise in the signal of the response. Correlation coefficients (Pearson's $r$) of decadal running mean AMOC strength and Denmark Strait (DS) overflow potential density ($p_{ref} = 0$) as well as March-mean mixed-layer depths (MLD) in the Labrador Sea. Correlations are computed from the reference experiment over 100 and 62 years for the coupled and forced configurations, respectively, applying a 11-year rolling mean to highlight a linkage on decadal and longer time scales. All values given in Sverdrup (1 Sv = $10^6$ m$^3$/s), except for the correlation coefficients in the last two columns.

| model configuration | mean state (Sv) | standard deviation monthly data (Sv) | mean response (Sv) | standard deviation of monthly response (Sv) | temporal correlation with | |
| --- | --- | --- | --- | --- | --- | --- |
| | | | | | DS overflow | Lab.Sea MLD |
| coupled | 16.8 | 3.8 | -1.5 [-1.8] | 3.9 | 0.57 | 0.74 |
| coupled-nested | 21.7 | 4.3 | -2.3 [-1.5] | 4.3 | 0.63 | 0.57 |
| forced | 13.8 | 2.9 | -4.9 | 0.3 | 0.80 | 0.25 |
| forced-nested | 17.1 | 3.4 | -4.6 | 1.7 | 0.82 | 0.21 |

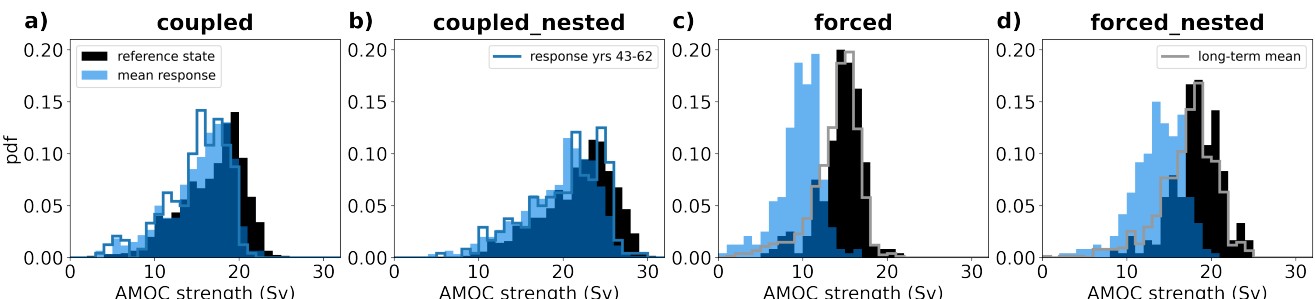

**Figure 4.** Statistical distribution of monthly-mean AMOC maximum strength (Sv) in model reference states (black) and perturbed states. Light blue outline in panels (a) and (b) shows response distribution for years 43–62 and light gray lines in panels (c) and (d) depicts the distribution of the long-term mean period (years 1–62).

In both, the coupled-nested and forced-nested configurations, the explicit simulation of mesoscale eddies increases magnitude and internal variability to the AMOC, which agrees well with similar comparisons (Hirschi et al., 2020; Biastoch et al., 2021). Our simulations also confirm Hirschi et al. (2020) in finding a systemtic increase in AMOC strength by coupling the same ocean model to an interactive atmosphere.

Variability is measured as monthly standard deviation of the maximum strength (Tab. 2) and the eddy-related increase amounts to 0.5 Sv in both, coupled and forced configurations. However, the interactive coupling with the atmosphere adds





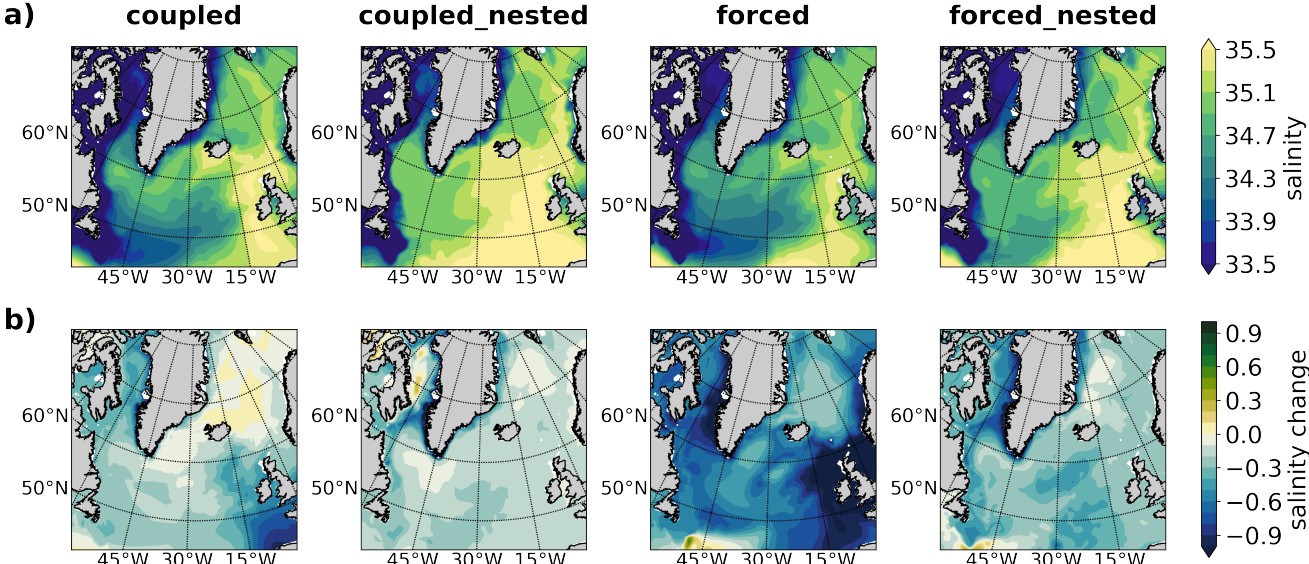

**Figure 5.** (a) Long-term mean state of upper ocean (0-200m) annual-mean salinity and (b) the change due to the freshwater perturbation. The response is computed as difference of the means over years 151–200 and 43–62 in the coupled and forced cases, respectively.

'noise', too, and AMOC variability is enhanced by about 1 Sv in the coupled setups compared to the respective forced ones. This is well depicted by the histograms of AMOC mean states in Figure 4 (black bars).

In response to the freshwater perturbation we expect the AMOC to weaken, which is simulated by all four model configurations. This is depicted by light blue histograms in Figure 4. The magnitude of the reduction in maximum overturning strength varies considerably. We find the two forced configurations much more sensitive simulating declines of 4.9 and 4.6 Sv with the overall weakest AMOC being most vulnerable. The coupled configurations simulate reductions of 1.5 to 2.3 Sv. These longer simulations also show that internal variability—and thus the period used as reference for computing the change—matters,

which can be seen in the difference between the two runs growing with extended perturbation. In fact, decadal mean AMOC weakening in the last 50 years of the perturbed run (i.e. 5 independent samples) ranges from 1.0 to 1.9 Sv for the coupled and varies within 1.1–3.0 Sv for the coupled-nested experiment.

The relatively wide distributions of AMOC strength seen in Figure 4 are due to both, interannual variability and a prominent annual cycle of about 7.1–8.5 Sv and 4.5–5.3 Sv in the two coupled and two forced reference experiments with systematically

larger amplitude in the eddying simulations. A significant change in seasonality in response to the freshwater perturbation is not found, however.

**Large-scale upper ocean salinity and freshening**

The additional freshwater released from Greenland enters the East and West Greenland Currents (EGC and WGC) and is transported south with the Labrador Current. Reaching Newfoundland, or rather the so-called Northwest Corner, we find

major deviations in its further path (see Fig. 5). Lack of mesoscale dynamics (being parameterized instead) and a resulting too



zonal positioning of the NAC lead to a massive, NAC-focused advection of freshwater across the Atlantic in the non-eddying simulations. In contrast, the strongly eddying simulations, which simulate narrower, stronger boundary currents and a more realistic Gulf Stream separation and Northwest Corner dynamics, accumulate freshwater along the American coast down to the Mid-Atlantic Bight. In these simulations, explicitly resolved eddies then entrain the freshwater into the Gulf Stream extension
and NAC, which is further discussed in Section 3.2.

The explicitly simulated eddies and stronger meandering cause a much wider spreading of the freshwater across the SPNA than achieved by the eddy parameterization in the coarse-resolution models, which are limited by their biased mean flow. This is well expressed in the change in upper ocean salinity, here averaged over the top 200 m, in response to the freshwater perturbation (Fig. 5b). While all four configurations yield an overall freshening of the SPNA, the upper ocean freshening is
stronger in the two non-eddying simulations including a significant intensification towards the east. This causes a massive freshening of >1 on the very eastern side of the SPNA, on the European shelf, with implications for regional sea-level change (not shown). Such piling-up of a fresh anomaly is not present in the eddying simulations. Instead, horizontal gradients are much weaker than in the non-eddying counterparts.

Similarly prominent is the stronger freshening of the forced experiments. This is an indirect effect in consequence of the
greater weakening of the AMOC in these simulations compared to the coupled ones as shown above. The reduced AMOC means less northward advection of saline and warm subtropical waters (see e.g. Smeed et al., 2018). As detailed by Griffies et al. (2009) and discussed further below, forced model configurations are prone to an overly strong positive salinity feedback. Other secondary processes causing an upper ocean salinity response are a southward advancement of the marginal sea-ice zone along the northern rim of the Labrador Sea and associated southward shifted melting of sea ice in all but the coupled
experiment, and a major decrease in deep mixing in the Nordic Seas in the forced experiments, of which notably only the non-eddying one shows a related freshening here (further discussed below).

**Large-scale upper ocean temperature and cooling**

In consequence of a weaker AMOC, the SPNA experiences wide-spread cooling. Averaged over the top 200 m representative for the upper ocean, this cooling can reach 2.1–2.3°C in some areas on annual mean (Fig. 6). SST exhibits an even larger decline
by 3.1–4.5°C, especially in mid-winter (February or March), where cooling occurs in the Labrador Sea due to reduced deep convection activity and a more extensive sea-ice coverage. This is except for the coupled experiment with smallest AMOC decline, in which annual mean cooling is limited to <1.3°C. All experiments also exhibit pronounced cooling in the eastern SPNA, which is as strong as—in coupled even stronger than—the temperature decline in the northern Labrador Sea. This eastern cooling is carried by the Irminger current from the ENA into the Irminger Sea.
The response is different and quite diverse for the Nordic Sea. We observe a warming of up to 2.3°C in the top 200 m in the coupled experiment but only <1°C in all other cases. The relatively strong warming in the coupled experiment is caused by an overall comparatively weaker advection of the freshwater perturbation into the Nordic Seas (Fig. 5b) so that the mixed layer is actually slightly deepening across the Nordic Seas (Fig. 7). The warming extends to 1000 m and is associated with a slight increase in the overturning north of the overflows. In the non-eddying forced experiment, the freshening in the upper 200 m
is much more pronounced, also in the Nordic Seas. In consequence, we find a similar warming pattern along the recirculation





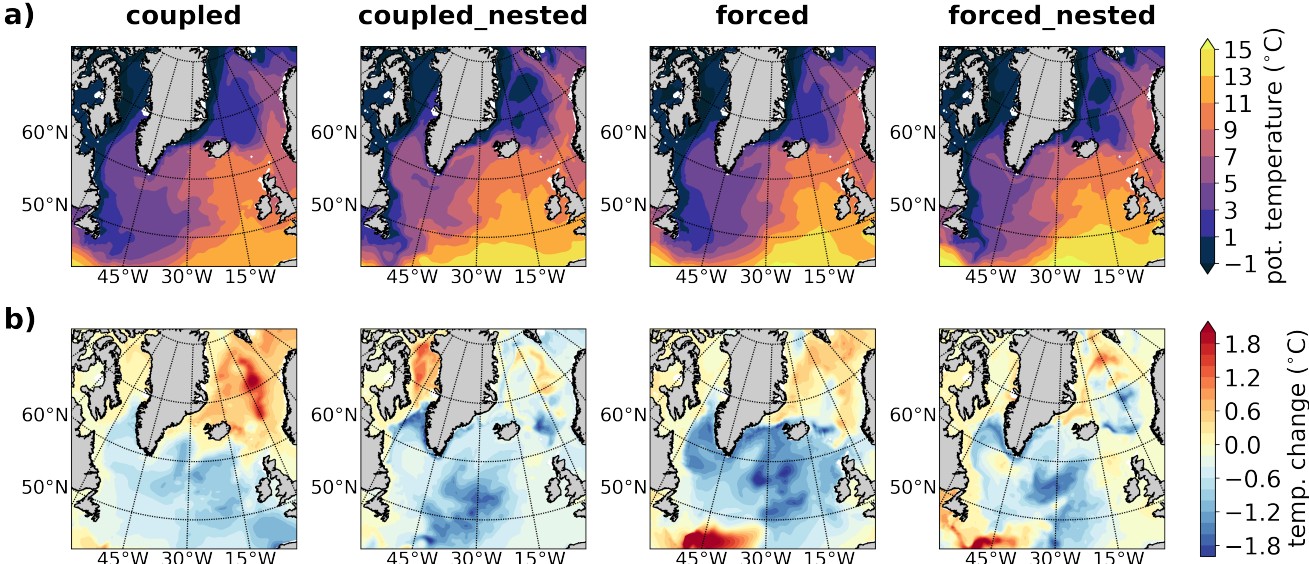

**Figure 6.** (a) Long-term mean state of upper ocean (0-200m) annual-mean potential temperature and (b) the change in consequence of the freshwater perturbation. The response is computed as difference of the means over years 151–200 and 43–62 in the coupled and forced cases, respectively.

path of the Atlantic water as in the coupled run, but the warming is about 0.5°C weaker as the mixed layer rather shoals. The two nested experiments both feature an overall stronger inflow of Atlantic water into the Nordic Seas and the freshening in the perturbation experiment is spatial much more evenly distributed (Fig. 5). There are only rather local regions of warming found in these two experiments. Instead, the cooling response found in the SPNA extends into the Nordic Seas (Fig. 6b).

**Watermass transformation**

For the illustration of the different watermasses and their transformation in the four model configurations as well as their property changes under the freshwater perturbation we define a number of regions following the path of the Atlantic water to the deep convection sights in the SPNA and Nordic Seas. For simplification, we only show the spatial means for these areas and averaged over the top 200 m if not mentioned otherwise. All averages are computed as grid-cell volume weighted means.

The regions of interest are depicted in Figure 8 with the color coding associated with the warm northward inflow on the eastern side (red-orange-yellow colors) and cold southward flow of Polar water (cyan) and deep convection regions in the west (blue and purple colors). This color coding is being used in subsequent plots of Figures 9 and 11. The eastern North Atlantic (ENA) is defined as the region between 15–30°W and 50–60°N (cf. Koul et al., 2020; Holliday et al., 2020). In addition, we provide mean potential temperature and salinity at the overflow depth of Denmark Strait (DS).

Focussing first on the mean states and beginning in the ENA (dark red filled circles) our simplified TS-diagrams clearly illustrate the strong fresh-bias present in the mean state of the two non-eddying simulations, in which ENA salinity is 0.8–0.9 lower than in the two nested, strongly eddying simulations (Fig. 9). Interestingly, taking the average just east of the ENA



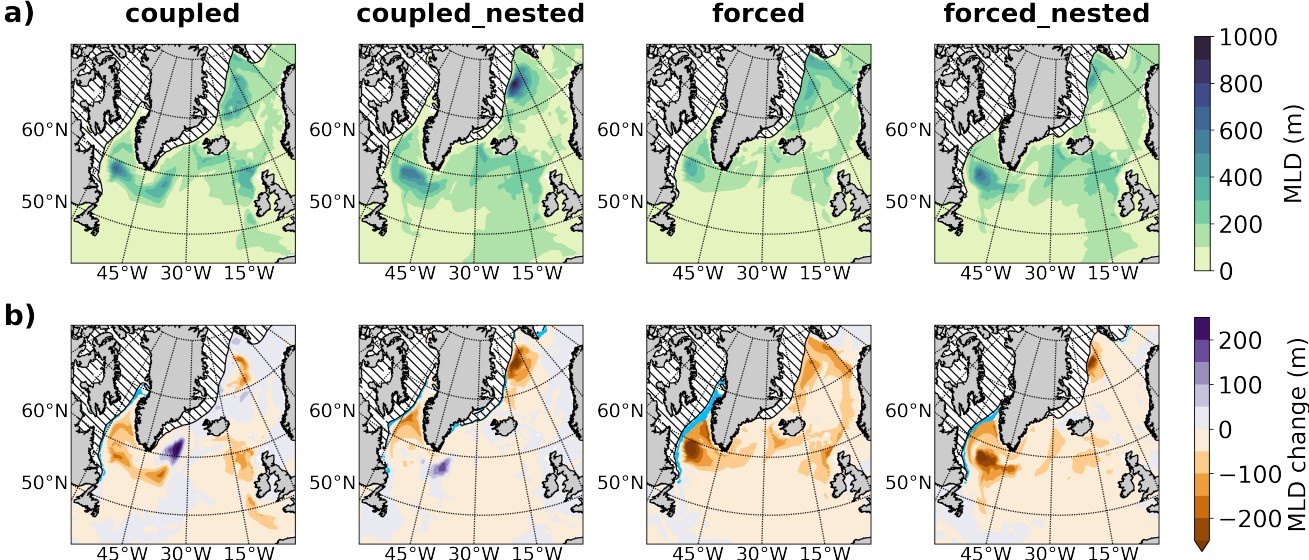

**Figure 7.** (a) Long-term mean state of annual-mean mixed-layer depth and (b) the change in consequence of the freshwater perturbation. The response is computed as difference of the means over years 151–200 and 43–62 in the coupled and forced cases, respectively. Areas covered by sea ice in the reference state are depicted in white, sea-ice expansion in the perturbed state is bright blue.

region, toward the European shelf (here referred to as ENA shelf, bright red) the salinity deviations are much smaller, within 0.1. In particular in the coupled-nested configuration, properties in the ENA and on the ENA shelf are almost identical—

considering the large scales we focus on. It is this temperature and salinity (TS) characteristic that sets the baseline for the watermass transformation in the SPNA and Nordic Seas. With 10–11°C the potential temperature is very similar for all model configurations. The coupled experiments must thus be considered relatively warm as these are conducted under pre-industrial conditions whereas the forced ones represent present day climate.

     In the TS-diagram, we can draw an almost straight line from the ENA shelf to the Norwegian Sea (orange), the central

Nordic Seas (yellow) and the Denmark Strait overflow (black). The line would have the approximately same slope for all model configurations. It would depict the freshening and predominantly cooling of the Atlantic water along its recirculation path though the Nordic Seas and deep convection. However, in the nested simulations the Nordic Seas convective mixing reaches deeper, exceeding 1500 m, than in their non-nested counterparts. Consequently, Denmark Strait overflow water remains 1.5–3°C warmer than in the eddy-parameterized runs. The deeper the winter mixed layer in the central Nordic Seas, the warmer

the overflow water.

     A source of strong freshening is the cold Arctic-sourced Polar water (cyan circles). Transported south in the EGC, it interacts south of Denmark Strait with waters of the Irminger Current carrying properties of the ENA (or rather ENA shelf) region. We find upper ocean properties of the Irminger (blue) and Labrador Seas (purple) situated roughly half way between these two source waters. The watermass in the interior Labrador Sea has a tendency to be slightly cooler and fresher than the one in





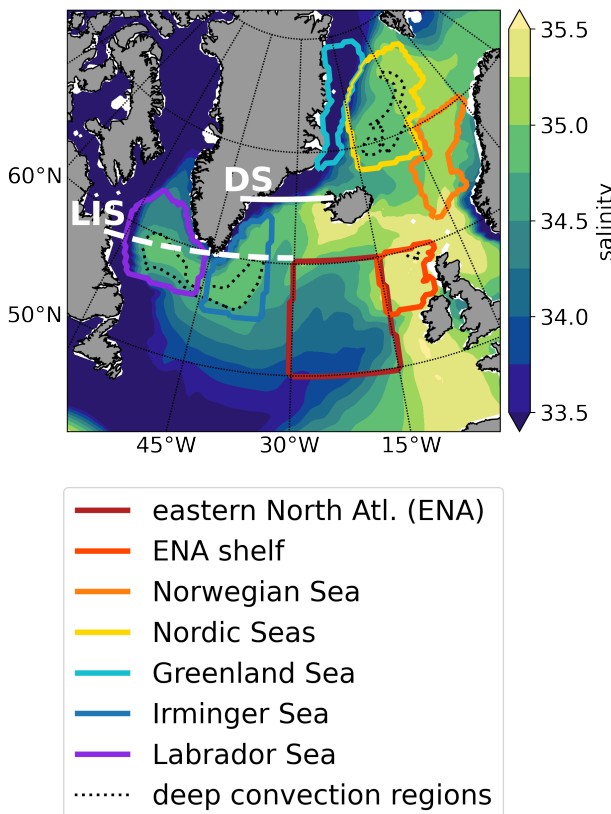

**Figure 8.** Map of regions selected for computing water properties and mixed-layer statistics. All regions are defined by geographic coordinates. Shelf regions are excluded by prescribing a minimum depth of 1500 m for Nordic, Irminger and Labrador Seas and of 1000 m for the Norwegian Sea. The ENA shelf region excludes areas shallower than 100 m. In contrast, the Greenland Sea region is defined for areas shallower than 500 m. White lines indicate the Denmark Strait (DS, solid) and Labrador/Irminger Sea (LIS, dashed) cross-sections. For reference, the long-term mean salinity of the coupled simulation averaged over the top 200 m is shown (cf. Fig. 5a) along with the March-mean 500-metre mixed-layer depth contour because these quantities guided the region selection. [Note, this is meant as single-column figure hence the legend with large font below the map.]

the Irminger Sea. Interestingly, the density difference between the two is a little larger in the eddying models, which resolve the boundary currents much better. While boundary current salinity is relatively similar among the simulations (small purple circle), the mixing between shelf and interior Labrador Sea are fundamentally different between eddy-parameterized and eddying models resulting in a greater salinity gradient across the shelf front in the two nested simulations. Moreover, the coupled runs exhibit a stronger salinity and thus density gradient across the shelf front than the forced configurations. We

identify deep convection and Ekman transport as likely causes. These will be discussed in more detail below and in section Section 3.3.




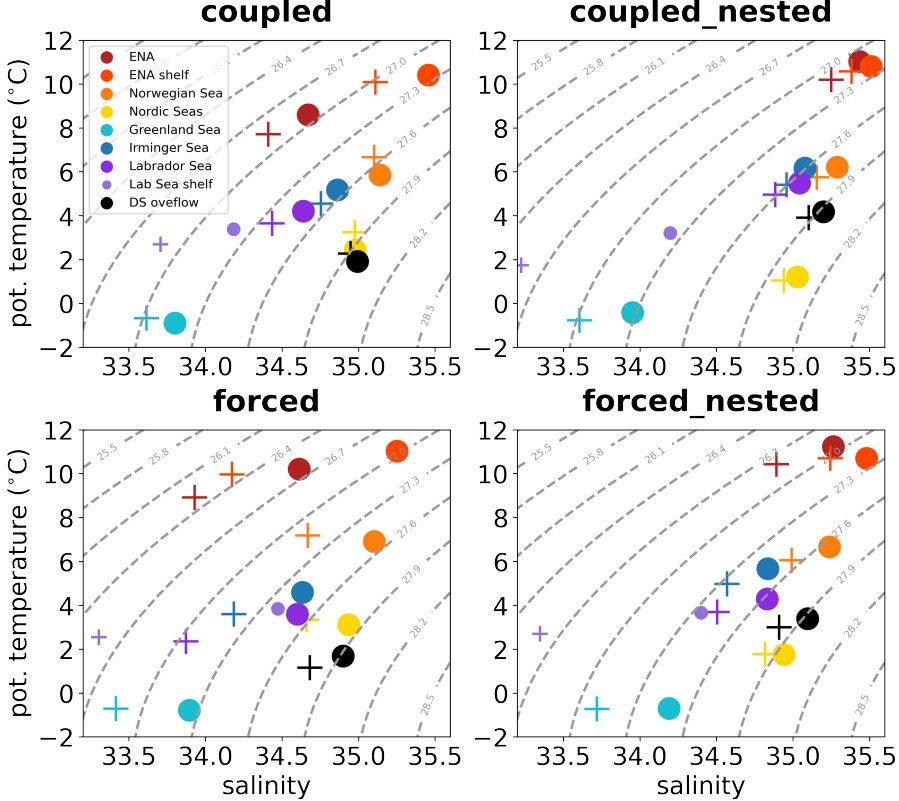

**Figure 9.** Mean water properties of regions depicted in Figure 8 (same color coding) from the reference simulation (filled circles) and the perturbation experiment ('plus' symbols). Symbols depict properties averaged over the upper 200 m, except for the Denmark Strait (DS) overflow (in black) where values from deepest grid cells (approx. 600 m depth) along the section at approx. 65.7°N are averaged. For the Labrador Sea region (purple) properties on the continental shelf (smaller symbols) are shown in addition to the ones from the interior part (regular sized symbols). The shelf is defined as areas shallower than 500 m within the same latitude bands as the deep, interior Labrador Sea (purple frame in Fig. 8).

The freshwater perturbation leads to a freshening and cooling in the ENA and on the ENA shelf in all configurations (compare crosses '+' with filled circles). The response is larger in the forced configurations than in the coupled ones and larger in the non-eddying than in the respective strongly eddying simulations. Particularly in the forced experiment this leads to a
density decrease of 0.4 kg/m³ whereas in the coupled-nested one there barely is a density change noticeable. In particular the freshening has consequences for the imaginary linear connection to the Nordic Seas in TS-space (connect red-orange-yellow-black crosses). The slope of this line steepens considerably with the northward decrease in salinity present in the reference state practically vanishing in the two non-eddying perturbation experiments. This is except for the forced run, where the ENA presents with excessive freshening.



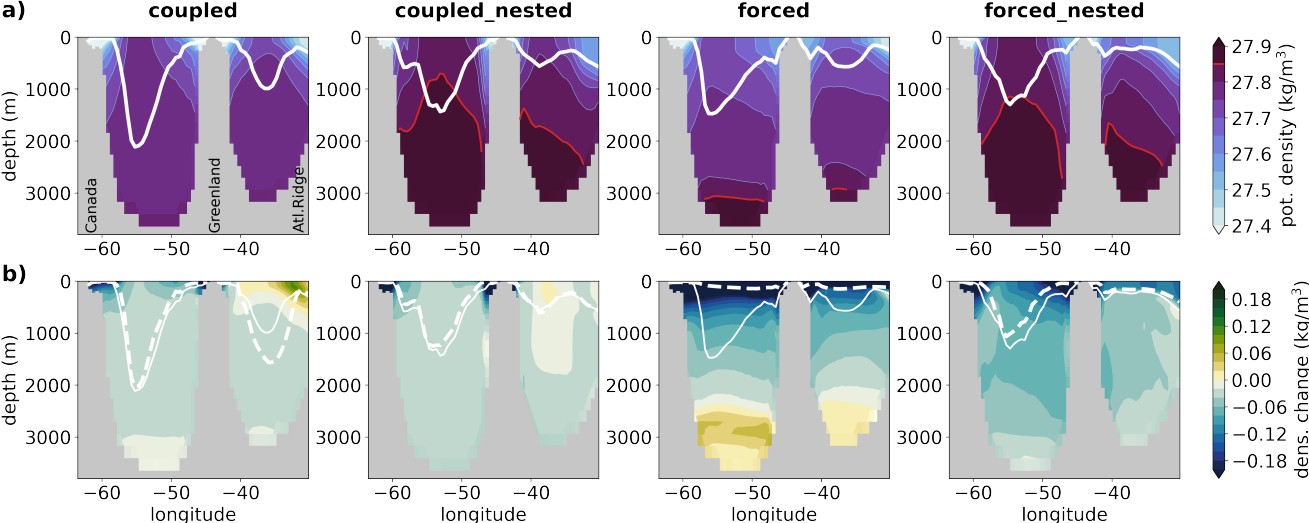

**Figure 10.** Cross-section through Labrador and Irminger Seas at approx. 60°N showing (a) March-mean potential density ($\sigma_0$) and (b) the change due to the freshwater perturbation. The white solid and dashed lines depict the March-mean mixed layer depth of the reference and perturbed states, respectively. The red contour in panels (a) highlights densities >27.85 kg/m$^3$.

With respect to the AMOC weakening discussed above, we stress that the density of the Denmark Strait overflow hardly changes in the perturbed coupled experiments. There occurs, however, a significant density decrease by 0.1–0.15 kg/m$^3$ in the two forced experiments. The latter two happen to be the model configurations with the overall stronger AMOC decline, which we suggest is related to the change in overflow properties. This is supported by a strong correlation on decadal time scales between AMOC strength and DS overflow density (~0.8). The correlation with Labrador Sea winter mixed-layer depth

is much weaker (~0.23, see Tab. 2), which agrees well with the results of Biastoch et al. (2021).

The freshening in the central Labrador and Irminger Seas follows a similar pattern with a stronger response in the forced and the non-eddying experiments compared to their respective counterparts. The response in temperature is roughly equal among all configurations for this particular region. Again, the forced models with the stronger freshening in the Labrador and Irminger Seas deep convection regions also show greater AMOC weakening. However, it does not become clear from these watermass

property changes why the coupled-nested configuration has a tendency for a stronger AMOC decline than the coupled one (2.3 vs. 1.5 Sv, cf. Tab. 2).

Lastly, we point out that the freshwater perturbation enhances the salinity and density discrepancy between Labrador Sea shelf and interior (compare smaller and larger purple circles and similarly the crosses in Fig. 9). While the freshening in the boundary current is similar for all experiments (except for the coupled case), the interior Labrador Sea freshens more in the

two forced cases. We believe that this is related to the Ekman transport in the WGC and will shed some more light on this in Section 3.3.





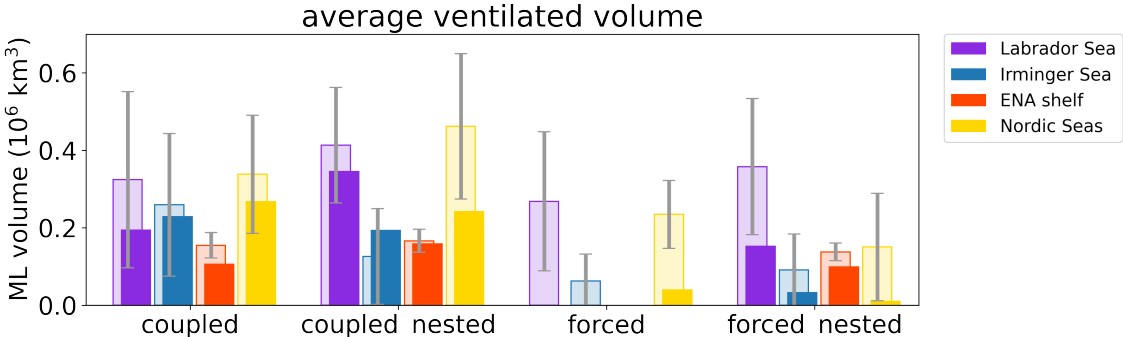

**Figure 11.** March-mean ventilated water volume for the reference state (pale bars in background) and perturbation experiment (in foreground) for regions with deep convection. Ventilated volume is computed as mixed layer depth (MLD) times grid-cell area where MLD>500m. Gray error bars indicate ± 1 standard deviation of inter-annual variability.

The density structure of cross-sections along ∼60°N through the Labrador and Irminger Seas presented in Figure 10 differs considerably between model configurations and has implications for the response to the freshwater perturbation. The watermass in the interior Labrador and Irminger Seas is denser in the two eddying configurations than in the non-eddying ones. Of the latter two, the forced configuration exhibits a stronger vertical density gradient. This is likely a consequence of the shallower deep convection in the forced configuration as well as the much shorter spinup (30 years) compared to the coupled experiment (1500 years) enabling properties of the initialization fields still visible in the deep ocean here. The non-eddying forced experiment features the smallest ventilated volume in the Labrador and Irminger Seas (Fig. 11, purple and blue bars). We define ventilated volume as mixed-later depth (MLD) times grid-cell area where MLD exceeds 500 m to focus on deep overturning. The larger the ventilated volume, the closer dense overflow waters rise to a mid-depth of approx. 1000 m. Here, overflow waters are identified by a density $\sigma_0$ >27.85 kg/m$^3$ (red contour in 10a) having been diluted along the way from ≥27.9 kg/m$^3$ at Denmark Strait (black circles in Fig. 9). In particular the eddying configurations show large content of overflow water in the deep Labrador and Irminger Seas. In contrast, the non-eddying coupled simulation with the densest overflow water at Denmark Strait shows no water mass identifiable as such in the Labrador and Irminger Seas cross-section. Interestingly, this configuration yields the weakest correlation between AMOC strength and DS overflow density (Tab. 2).

We also note that the two eddying configurations show much steeper isopycnal slopes between the boundary currents (bright colors) and the interior (dark blue) in Figure 10a. As will be further discussed below this is related to the 1/2° models running with an eddy parameterization but the 1/10° ones without though the higher resolution is not quite sufficient to simulate the full eddy spectrum.

Greenland meltwater enters the boundary currents enhancing the density gradient to the interior Labrador and Irminger Seas. Larger eddies spawned off the WGC due to unique topography around Cape Desolation and smaller, local ones created by baroclinic instability act to reduce this gradient (Rieck et al., 2019). The magnitude of freshening of the interior Labrador and Irminger Seas in the perturbation experiments are key to understanding the AMOC response. It is the forced non-eddying



experiment with the smallest difference between Labrador Sea shelf and interior water properties (Fig. 9) that presents with a
complete loss of ventilated volume for areas of deep convection (MLD>500 m) under the freshwater perturbation as shown in
Figure 11. This configuration features the freshest mean state (not shown) and therefore has the lowest barrier for any additional
freshwater carried by the EGC and WGC to enter the deep convection sites. With deep mixing being shut off, most meltwater
stays in the upper subpolar gyre (Fig. 12) further enforcing the response, which ultimate leads to the most pronounced AMOC
decline of almost 5 Sv among all configurations (Fig. 4 and Tab. 2). Eventually, this even leads to an accumulation of saltier
overflow water at depths >2500 m in the Labrador Sea and a warming between 100 and 1000 m (not shwon). As another
consequence, the least meltwater tracer is found to leave the SPNA with the deep water in this configuration (Fig. 13).

The other extreme is the coupled non-eddying experiment. While similar amounts of meltwater enter the upper interior
Labrador Sea (Fig. 13), the entire water column is more saline. In addition, deep mixing is stronger than in the forced counter-
part with the Irminger Sea adding considerably to the ventilated volume (Fig. 11). As a result, the depth of the winter mixed
layer does not change—or is compensated by deepening in the Irminger Sea, and deep water formation is less affected by
the freshwater perturbation. In general, deep convective mixing brings more heat and salt into the upper ocean in the coupled
configurations since there is a non-negligible heat (and salt) bias in these (Matthes et al., 2020). This helps to maintain the
overturning against the freshwater perturbation and in consequence, AMOC strength weakens only by 1.5 Sv.

The response in the two eddying simulations suffers from the lack of exchange between the boundary currents and interior
in the Labrador and Irminger Seas already mentioned. Here, the meltwater is rather entrained to greater depths through mixing
across the SPNA, which is enhanced compared to the non-eddying configurations (Fig. 7)a. Ventilated volume in the Nordic
Seas decreases significantly in both experiments. With a generally saltier and warmer ocean, however, the AMOC of the cou-
pled configuration responds less strongly and the two nested configurations rather group with their non-eddying counterparts
than separating into an own category (Tab. 2).

## 3.2 Mesoscale ocean dynamics

In this section we discuss the outcome of explicitly simulating mesoscale eddies in the nested configurations instead of param-
eterizing their effect. We focus on regions and processes of particular relevance for the distribution of Greenland meltwater and
its potential impact on deep water formation. This is exchange between the boundary current and interior in the Labrador Sea,
entrainment into the North Atlantic Current and leakage into the subtropical gyre.

**Boundary currents**

The ocean-grid refinement improves the dynamics in the nest region significantly (Fig. 1a). We find a strongly eddying ocean
where the 1/10° grid sufficiently resolves the Rossby radius, which is the case south of approximately 50°N. In higher latitudes
the finer resolution yields stronger and more focused boundary currents, such as in the Nordic Seas and the Labrador as well
as Irminger Sea, but is inadequate to simulate the full dynamical mesoscale spectrum. For example, the western boundary
current transport in the Labrador Sea at 53°N of the coupled model amounts to 33 Sv and that of coupled-nested to 53 Sv,
which is much closer to observations. We also find individual WGC eddies—or Irminger Rings—entering the interior northern
Labrador Sea (not shown). But smaller mesoscale eddies crucial for restratification after deep convection in winter (e.g. Rieck



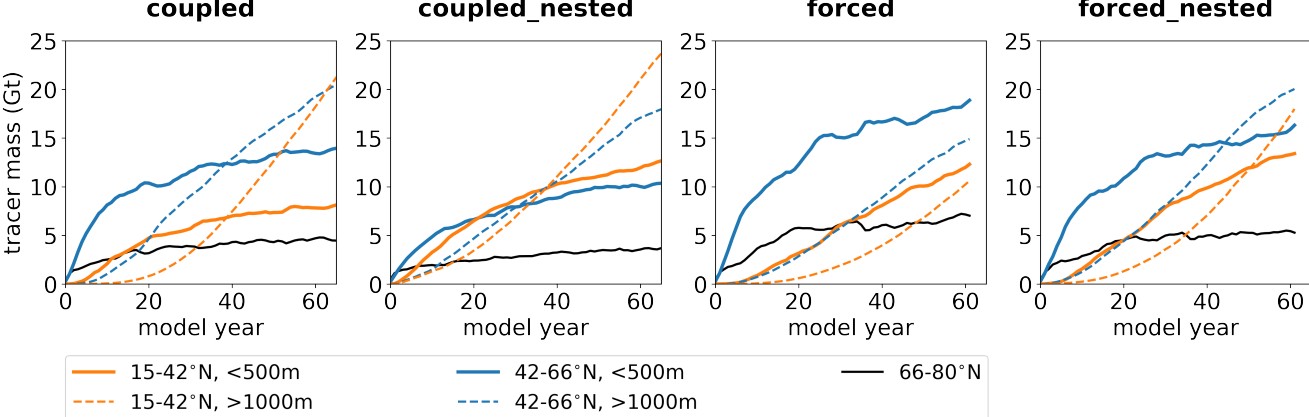

**Figure 12.** Total meltwater-tracer mass for selected regions of the North Atlantic integrated over the entire basin width for the upper 500 m (solid lines) and below 1000 m (dashed lines). Blue lines depict the tracer mass in the subpolar gyre between 42° and 66°N , orange lines in the subtropical gyre (15–42°N). The black lines combine Nordic Seas and Baffin Bay tracer-mass inventory, which are approximately of same magnitude.

et al., 2019) are not explicitly simulated by the nested configurations. This results in generally steeper isopycnals seperating the Labrador Sea shelf from the interior (Fig. 10a). Here, the models parameterizing the cross-frontal transport by eddies show
a more realistic hydrography. We suspect that this is an important factor for the overestimation of deep mixing already in the mean state of the nested simulations (Fig. 11) and an underestimation of Greenland freshwater tracer concentration in western subpolar gyre in the nested perturbation experiments (Fig. 13). Due to the latter, our results appear contradictory to those of Böning et al. (2016) at first glance but in fact simply highlight the necessity of using at least 1/20° grid resolution to properly simulate the mesoscale in this region. This is further discussed in Section 4.

**The Northwest Corner**

The Northwest Corner off Newfoundland is a dynamically highly active region where the southward flowing Labrador Current carrying fresh and cold Polar waters meets the North Atlantic Current (NAC) transporting warm, salty subtropical water northward. At this switchyard mesoscale ocean dynamics determine how much Polar water—and hence Greenland freshwater—is either mixed into the NAC or continues travelling south into the Mid Atlantic Bight (Böning et al., 2016;
New et al., 2021). Here, we find major divergence between the non-eddying and strongly eddying model configurations. The refined grid of the nests do not only yield a largely improved North Atlantic cold bias but also a much more realistic, reduced entrainment of Greenland-sourced freshwater into the NAC. As in the study of Böning et al. (2016), our strongly eddying simulations transport significantly more meltwater along the coast all the way into the Mid Atlantic Bight than the non-eddying models. The eddy parameterization used in the latter prescribes a too strong entrainment into the NAC at the Northwest Corner,
which is well seen in the tracer concentration of the upper ocean (Fig. 13, at 50 m depth). Differences in the wind field and exact position of the NAC may explain the elevated tracer concentrations in the forced compared to the respective coupled



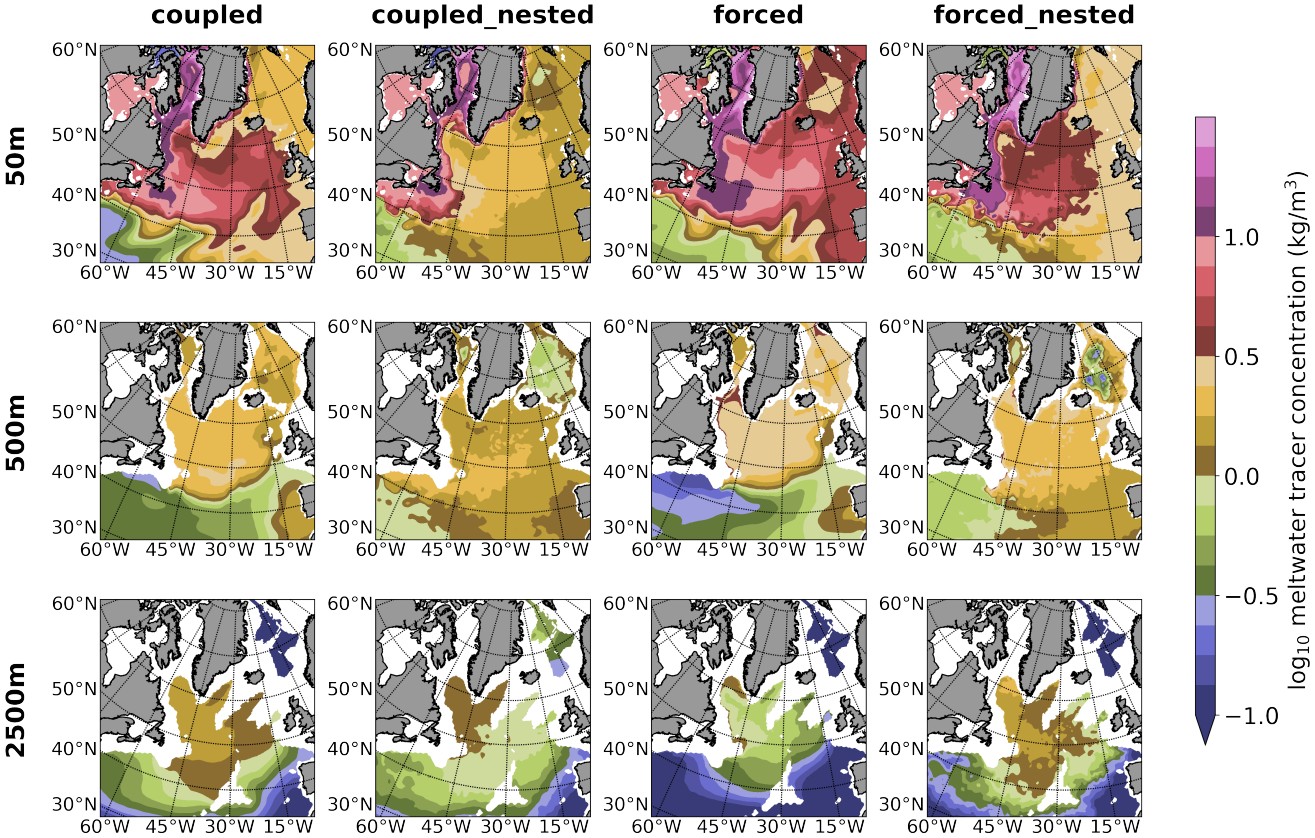

**Figure 13.** Snap shots (5-day mean) of meltwater passive tracer concentration at three depth levels after 50 years of simulation with freshwater perturbation.

experiments in this region. We find imprints of the different Northwest Corner dynamics on meltwater tracer concentrations as deep as about 1000 m.

### Gyre-gyre exchange

Not only the meltwater concentration in the NAC itself is affected by resolving the mesoscale but also the cross-frontal exchange between subpolar and subtropical gyres along the NAC axis. In the strongly eddying simulations the NAC consists of a wide field of meanders and eddies whereas it resembles a laminar flow associated with clearly defined fronts in the non-eddying configurations. The dynamically rich eddy field does not only yield a much more homogeneous distribution of the meltwater across the SPNA but also an enhanced exchange with the subtropical gyre. This exchange already occurs far west in

the Gulf Stream extension and persists across the entire Atlantic basin width. For example, this is visible in cold-core eddies carrying meltwater tracer across the front of the Gulf Stream and NAC (Fig. 1b). This mixing is extending over the entire upper and mid-depth ocean, at 50 as well as 500 m depth (Fig. 13). Therefore, leakage of meltwater into the subtropical gyre is stronger with explicitly simulated eddies than with the typical eddy parameterization. Figure 12 shows an earlier occurrence





and faster growth of meltwater-tracer mass in the subtropical gyre for both the upper ocean above 500 m depth and for the deep
ocean below 1000 m. The contrast is particularly strong between the coupled experiments. Here, the non-eddying simulation
maintains twice as much tracer in the upper 500 m of the subpolar than in the subtropical gyre after 50 years of continued
freshwater perturbation whereas the tracer content of the subtropical gyre exceeds the one in the subpolar latitudes after 20
years in the strongly eddying experiment. The divergence is smaller between the two forced configuration. We suspect different
wind forcing and a stronger meridional density gradient in the NAC region to play a role in this weaker contrast among the
forced than the coupled configurations.

## 3.3 Atmospheric coupling

Lastly, we address the relevance of ocean-atmosphere coupling for Greenland freshwater-release experiments by comparing
the two coupled with the respective forced simulations. The stark contrast in the AMOC response between these two sets of
configurations (see Tab. 2) suggests that atmosphere-ocean fluxes play a major role. Coupled models are likely less sensitive to
a perturbation such as the additional freshwater inserted here. This is because changes in AMOC strength are associated with
both, a positive salinity and a compensating negative temperature feedback with the latter only being active if the atmosphere
is able to respond to changes in SST like in a coupled model configuration (Griffies et al., 2009). However, coupled modeling
may have its downside in case the atmospheric fluxes are biased as the coupled model has more freedom. In the following we
address both topics using the surface heat flux and coastal winds to discuss large-scale and regional effects of ocean-atmosphere
coupling and its impact on freshwater-release experiments off Greenland.

### Surface heat flux

We focus our analysis of the surface heat flux (SHF) on the winter season, averaging from December to February (DJF),
which dominates the annual mean. Heat loss from the ocean to the atmosphere across the SPNA during fall and winter increases
upper ocean density and hence plays an eminent role in preconditioning the ocean for deep convection with a peak in March.
All four model configurations show similar patterns and magnitudes of SHF in their mean states (Figure 14a). A maximum heat
loss, which exceeds $400\,\mathrm{W/m^2}$ is located along the northwestern side of the Labrador Sea along the sea-ice edge, which expands
from Davis Strait along the Canadian shelf. There also is pronounced heat loss in the northeastern part of the SPNA (southeast
of Iceland as well as over the Irminger Current) and over the boundary currents in the Nordic Seas and around Greenland.
The block-like structure in the SHF output of the coupled configurations is due to the surface fluxes being computed on the
coarser grid of the atmospheric model at a horizontal resolution of about $1.9°$ (but stored on the ocean model grid at $0.5°$).
While the large-scale pattern is very similar in all reference simulations, we note a difference in the region of the Northwest
Corner, east of Newfoundland. In the non-eddying configurations, which feature a strong SST cold bias in this region (cf. Fig.
3), a near-zero SHF can be found. In contrast, the coupled-nested configurations, almost completely lacking this bias, simulates
a heat loss of around $200–250\,\mathrm{W/m^2}$ in this region. However, this has little effect on downstream upper ocean temperatures,
which are in the ENA (and ENA shelf) region roughly all the same across the different configurations (Fig. 9).

In the coupled experiments the atmosphere can adjust to changing ocean surface conditions. In contrast, the atmosphere
may act as an infinite source (or sink) of energy to the ocean in the forced experiments. This is well expressed in the DJF-




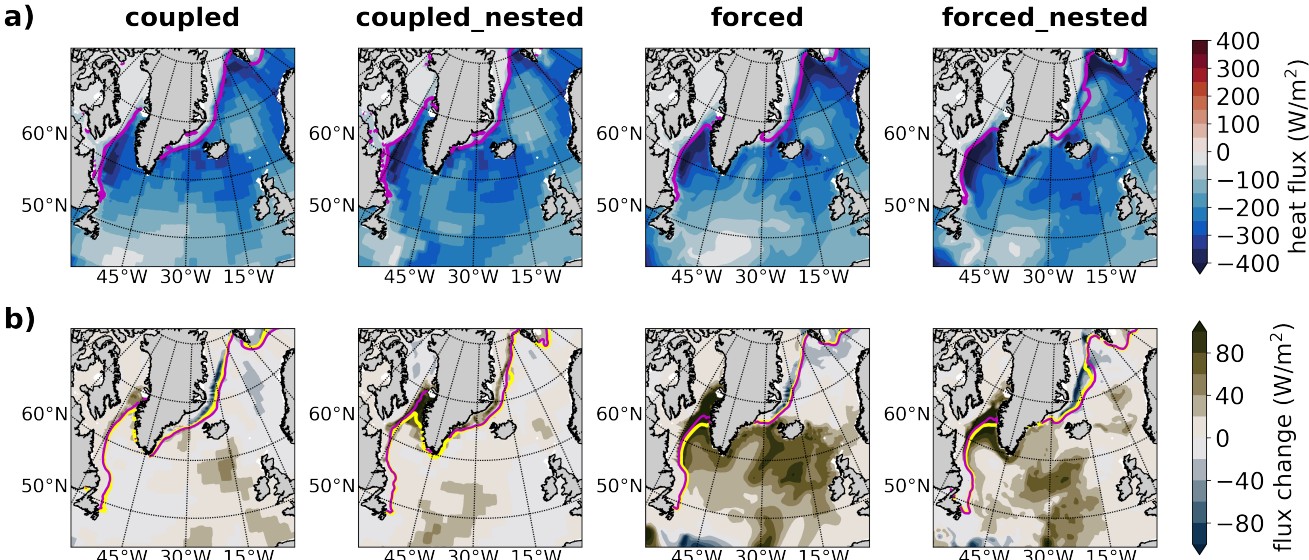

**Figure 14.** (a) Long-term mean surface heat flux (negative is ocean heat loss) )averaged over December to February (DJF). (b) Response in surface heat flux (positive means reduced heat loss) to freshwater perturbation, DJF mean. Magenta and yellow lines depict the sea-ice edge (15% ice concentration contour) for the reference and perturbed states, respectively.

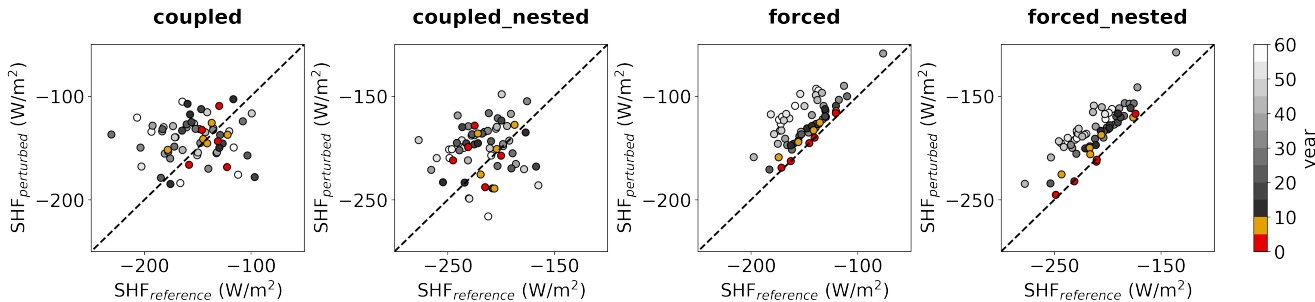

**Figure 15.** Changes in surface heat flux (SHF) due to the freshwater perturbation. SHF is averaged over the eastern SPNA (35–15°W and 45–60°N ) and over December to February (DJF). Circles for each of the first 62 years of the perturbed simulations vs. the respective years of the reference runs are displayed. Years 1–5 and 6–10 are highlighted in red and yellow respectively. Axis scales are shifted by -50 W/m$^2$ for the two nested configurations. The dashed line depicts the 1:1 line.

mean SHF change presented in Figure 14b. The two coupled experiments yield SHF changes of less than 20 W/m$^2$ in most of the SPNA and nowhere more than 50 W/m$^2$. The forced experiments, however, present with a distinct pattern of SHF
increase, which here means ocean heat loss decrease. The pattern resembles very much the cooling pattern of upper ocean temperature shown in Figure 6b. Without an adjustment in atmospheric temperatures, the upper ocean cooling in response to the freshwater perturbation and AMOC weakening reduces the temperature difference between ocean and wintery cold





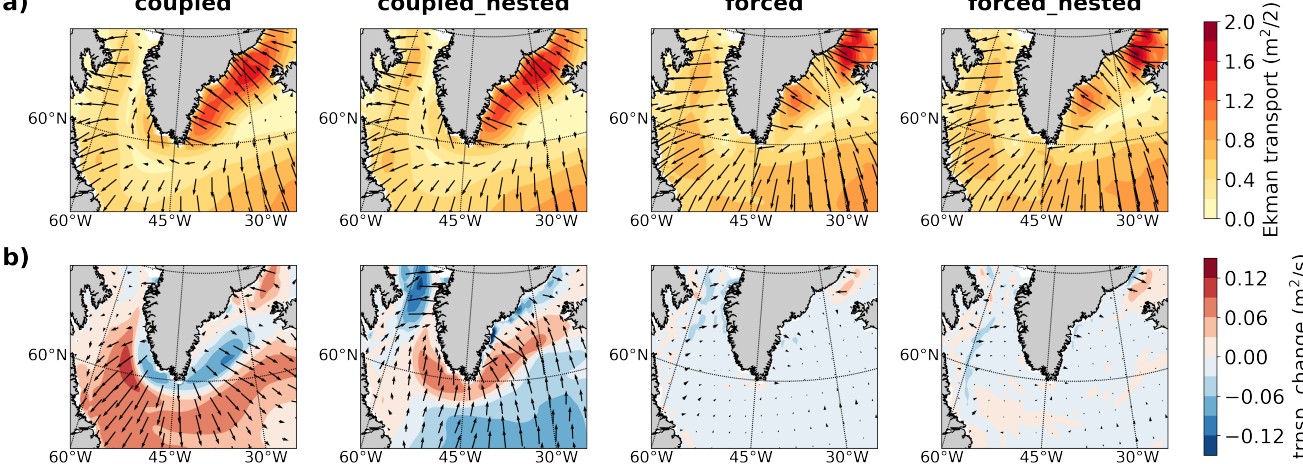

**Figure 16.** (a) Long-term annual mean Ekman transport magnitude (colored contours) and direction (arrows, length relates to magnitude). (b) Response in Ekman transport to freshwater perturbation.

atmosphere and thus yields a reduced heat loss of up to 100 W/m², over the entire but mostly eastern SPNA. This reduction evolves rather slowly with the decline of the AMOC and is negligible during the first 5 years after onset of the perturbation

but grows to 20–40 W/m² in the eastern SPNA during the first decade (not shown). AMOC weakening yields a comparable response in the coupled experiments during the first 5–10 years, which however is masked by internal variability of similar or larger magnitude and by the damping effect of the coupled atmosphere. The SHF difference between perturbed and reference run of the forced simulations remains positive (i.e. in a state of reduced heat loss) continuously throughout the experiment, whereas in the coupled system internal variability and the interactive atmosphere drive differences of opposing sign (Fig. 15).

In the coupled-nested configuration differences are skewed towards positive values, which may explain or contribute to the slightly stronger AMOC response on the long run (2.3 vs. 1.5 Sv, see Tab. 2).

We also note a consistently larger heat loss by the two strongly eddying configurations compared to the non-eddying ones ranging from -250 to -150 W/m² rather than -200 to -100 W/m² (Fig. 15), which we attribute to sharper SST gradients from mesoscale filaments, a generally wider NAC and a stronger AMOC in the nested configurations (cf. Fig. 14a). The positive

SHF anomalies in the freshwater perturbation experiments (14b) are tightly associated with a reduction in the upward surface freshwater flux (evaporation minus precipitation, not shown). This provides another positive feedback favoring further AMOC weakening in particular in the forced experiments.

**Ekman transport**

The Greenland high and Iceland low pressure systems create predominant north-easterly winds over the continental shelf in

the Irminger Sea. As a result onshore, downwelling favorable Ekman transport (Fig. 16a ) confines freshwater carried with the EGC (and its coastal sibling, the EGCC) to the shelf and shelf break (Fig. 5a). This is present in the atmospheric fields applied to the forced experiments as well as in the coupled configuration with the latter showing 30-40% larger Ekman transport





magnitudes. Over the WGC region this is different. The coupled atmosphere tends to extend the low pressure regime into the Labrador Sea creating a similar gradient toward the Greenland high, which drives onshore Ekman transport here as well. In contrast, the forced simulations feature offshore Ekman transport pushing fresh Polar waters away from the shelf. We consider this a major cause for finding a generally fresher upper ocean in the central Labrador Sea of the two forced experiments and much fresher conditions on the Labrador Sea shelf of the coupled simulations (Fig. 9).

While there is only negligible wind stress change in the two forced experiments, both coupled experiments yield a slight weakening of the Icelandic low showing sea-level pressure increases of 0.5-1 hPa over the Labrador and Irminger Seas with a stronger increase in the non-eddying experiment. In the latter this results in a weakening of the onshore Ekman transport in the WGC and Irminger Sea regions (Fig. 16b). Interestingly, the opposite is the case in the coupled-nested experiment, which may be related to the southward expansion of the sea-ice edge along the west coast of Greenland only showing in coupled-nested but not coupled experiments (cf. Fig. 14b). The particular reinforcement of the onshore Ekman transport over the WGC very likely keeps the coupled-nested configuration from yielding a larger freshening response in the interior Labrador Sea though it features the freshest boundary current (Fig. 9). The slackening of the Ekman transport in the non-eddying coupled experiment may facilitate the relatively similar freshening of interior and shelf regions found for this configuration (compare distances of purple circles and crosses in Fig. 9).

Summarizing, there certainly is a non-negligible influence on wind stress by an interactive atmosphere, which seems to reside less in a large-scale response to the broad surface cooling of the SPNA but rather in changes or difference in critical locations. In our case, this difference is most crucial for the mean state of the model and has only indirect influence on the response to the freshwater perturbation. Regarding the impact of the negative temperature feedback associated with AMOC changes missing in the forced experiments, we consider the damping of the surface heat and freshwater flux feedbacks in the coupled experiments a major reason for their weaker AMOC response. However, we cannot rule out an important influence by the different ocean mean states at the beginning of coupled and forced experiments as well.

## 4 Discussion

As the mass balance of the Greenland ice sheet grows increasingly negative the impact thereof on the ocean has become a major topic for the SPNA community. The meltwater is difficult to detect by oceanographic observations and hence has lead to an enhanced interest in numerical model simulations to project the near-term implications. While the potential of GrIS mass loss making a significant long-term and large-scale impact on ocean circulation and sea level has been explored in many model studies before—like we did in Martin et al. (2022), the subtle beginnings and regional effects as they may have occurred over the passed decade already, likely require very sophisticated, high-resolution, strongly eddying and even coupled models. This is what we argue for in the following. We also keep the focus to the SPNA as this is where we can expect changes to show first.

To set the stage we use the recently observed shift in deep convection from the Labrador to the Irminger Sea as an example (de Jong et al., 2018; Zunino et al., 2020; Rühs et al., 2021). Rühs et al. (2021) argued for a potential role of enhanced Greenland runoff in this shift and showed that salinity anomalies in the northern part of the Labrador Sea correlate well with those in the



EGC, in particular its coastal branch, which carries most of the runoff. Strongly eddying models have been used to show that WGC eddies are essential for this connection (Böning et al., 2016; Castelao et al., 2019; Georgiou et al., 2019). According to the model analysis of Rühs et al. (2021), the recent extreme freshening of the eastern North Atlantic (Holliday et al., 2020) arrives a little farther south, to the central Labrador Sea beign advected with the Irminger current located slightly further
offshore. Again (sub-)mesoscale processes play a role for connecting boundary current, deep convection and downwelling (Georgiou et al., 2019; Tagklis et al., 2020). Recent freshening by polar waters, runoff and Atlantic water salinity anomalies may thus have contributed to hindering deep convection in the northwestwern Labrador Sea. Potentially in consequence thereof, enhanced deep convection in the Irminger Sea has certainly offset any impact of recently enhanced runoff from Greenland on deep water formation. In this respect it is an intriguing feature of both of our coupled simulations that deep convection expands
considerably southeast and east of Cape Farewell into the Irminger Sea (Fig. 7b). We do not find such response in the forced experiments. In contrast to forced simulations by Böning et al. (2016) and Rühs et al. (2021), the atmospheric forcing we apply does neither cover the recent increase in Greenland runoff nor match the applied freshwater perturbation in time.

**Atmospheric coupling**

With respect to the role of atmospheric feedbacks and forcing, our experiments are tailored towards the objective whether
forced ocean (and sea-ice) simulations can be used to demonstrate and diagnose the impact of enhanced GrIS melting in the ocean. But the question regarding the influence of ocean-atmsophere interaction is not that simply answered.

On the one hand, our results show the strong influence of a missing negative temperature feedback for stabilizing the AMOC in forced experiments (c.f. Rahmstorf and Willebrand, 1995; Gerdes et al., 2006; Griffies et al., 2009), in which the AMOC weakens twice as much as in the coupled experiments (Tab. 2). This is despite differences in climate mean state (pre-industrial
for coupled vs. present in forced experiments), spinup length (1500 vs. 30 years, i.e. separation from observation-based initial ocean status), and surface salinity restoring (none vs. weak). The coupled model develops a quite typical negative/positive salinity bias above/below 600 m during the extended spinup (Matthes et al., 2020, their Fig. 14). This may make the coupled ocean also less susceptible to the prescribed moderate freshwater perturbation and support stronger deep convection once triggered. Further, the change in winter surface heat loss is less than 10% in the coupled experiments (Fig. 14), which may
lead to doubting the importance of a positive feedback related to temperature-adjusted atmosphere-ocean fluxes and rather give preference to a significant influence by the ocean mean state.

On the other hand, to unambiguously show the influence of ocean-atmosphere interaction on the response of the climate system to such enhanced freshwater input, the forced experiments would need to be conducted using the surface fluxes of the coupled control experiment as has been done by Stammer et al. (2011). Interestingly, our results oppose those of Stammer et al.
(2011) though the arguments are similar: In our setup the coupled configuration is less sensitive to the freshwater perturbation than the forced one. We see a similar causality of enhanced heat loss associated with enhanced precipitation over the SPNA supporting a further weakening of the AMOC just like Stammer et al. (2011) but in the forced instead of the coupled experiment. While upper ocean cooling and freshening in their forced ocean-only experiment is considerably weaker than in their coupled one, larger changes in surface fluxes over the SPNA in our forced experiment also drive greater cooling and freshening
compared to our coupled one (Figures 6b and 5b). We suspect that this opposing outcome is due to computing the surface fluxes





in the ocean model prescribing atmosphere temperature and winds from an unrelated reanalysis rather than prescribing surface fluxes of a related unperturbed coupled simulation. This may seem trivial but is an important argument for taking atmosphere feedbacks into account when quantifying the impact of current and future increases of freshwater input to the SPNA using models.

For studying and projecting the impact by GrIS mass loss using a coupled climate or ocean-only model, the representation of the coastal winds around Greenland will have major implications on the results of the simulation, too. As shown here but also by Castelao et al. (2019) and Duyck et al. (2022) upwelling favorable Ekman transport plays a significant role in spreading relatively fresh coastal waters offshore into the Labrador and Irminger Seas. Forcing an ocean model with reanalysis winds may thus yield a better representation of the Ekman transport, an issue that may be overcome by coupling with a high-resolution
atmosphere component.

### Role of mesoscale eddies

    Böning et al. (2016) investigated spreading of Greenland meltwater in an eddy-permitting (1/4°) and a strongly eddying (1/20°) ocean-only simulation and found that significantly more meltwater entered and accumulated also earlier in the central Labrador Sea in the strongly eddying model. At first glance our results seem contradictory with greater freshening (Fig. 5)
and higher meltwater concentrations (Fig. 13) in this region found in the non-eddying models presented here. However, the non-eddying model presented here uses the GM-parameterization to account for the effect of missing eddies, whereas the 1/4° model of Böning et al. (2016) did not include such parameterization. Further, our eddying model using a grid resolution of 1/10° (compared to 1/20° in Böning et al., 2016) only features larger eddies, so-called Irminger Rings or WGC eddies, which carry relatively fresh water from the boundary current into the interior Labrador Sea and therefore play a role in precon-
ditioning of deep convection, but not the many smaller ones, which are also crucial for the restratification process after deep convection in winter (Rieck et al., 2019). The eddies resolved in our model are obviously not sufficient for bringing enough meltwater to the deep convention sites to achieve results comparable to Böning et al. (2016). In a similar comparison of 1° and 1/10° model configurations, Weijer et al. (2012) already found that stronger boundary currents keep the freshwater anomaly away from the deep convection sites in the eddying model. Dukhovskoy et al. (2016) also highlight the importance of eddy
fluxes for Greenland meltwater runoff to enter the interior Labrador Sea in a model intercomparison involving similar ocean grid resolutions. This observation gains relevance as the global climate modeling community of CMIP increasingly employs grid resolutions of 1/4° –1/12° without eddy parameterizations, which is insufficient for regions of, for instance, deep and bottom water formation. We consider implementation of scale-aware eddy parameterizations such as proposed by Jansen et al. (2019) a promising solution for future application in high-resolution but not quite Rossby-radius resolving models.

Biastoch et al. (2021) and Yeager et al. (2021) noted the importance of explicitly simulating mesoscale dynamics for an improved representation of the transport of Denmark Strait overflow water and the maintenance of its characteristics along the way for forced ocean-only and coupled climate model simulations, respectively. Our results clearly support this and—as we a apply a relatively coarse vertical grid with only 46 levels—stress the relevance of more accurately simulating the boundary current in addition to the overflow itself for reducing dilution of this crucial water mass on its way to the Labrador Sea. In
addition we note, that our forced experiments show a higher correlation between AMOC strength and the overflow than the





coupled ones though the latter tend to have a slightly greater overflow water density. With respect to the above discussion, we speculate that the atmospheric fluxes have significant influence on this. Likely also the much shorter spinup of the forced experiments play a role. After more than 1500 years the coupled experiments have certainly drifted away from the observed initialization state whereas the initialization may still have a beneficial effect on the forced configurations.

Our results are in line with the notion that explicitly simulating mesoscale eddies—rather than parameterizing their effect— does not significantly impact the response of the AMOC to global warming or freshwater anomalies. This specifically holds for the magnitude of the response and less for the adjustment time scale (similar to Weijer et al., 2012). Winton et al. (2014) noted that AMOC state and variability are more important for the model's sensitivity than grid resolution. And Gent (2018) summarizes studies comparing non-eddying and eddying models for their AMOC sensitivity to buoyancy forcing and finds

neither clear evidence for the AMOC being more sensitive in strongly eddying models nor grid resolution a dominant factor. Hirschi et al. (2020) and Jüling et al. (2021) have recently added evidence along these lines. However, like earlier studies (e.g. Weijer et al., 2012; Böning et al., 2016; Jüling et al., 2021), our experiments show that redistribution of meltwater and response patterns to enhanced input thereof become significantly more realistic with increasingly strong eddying simulations. The realistic simulation of processes in the deep water formation regions require grid resolutions of $1/20°$ or finer to resolve

the necessary eddy spectrum (Böning et al., 2016; Rieck et al., 2019; Castelao et al., 2019; Georgiou et al., 2019; Tagklis et al., 2020; Pennelly and Myers, 2022). In turn, this suggests that the above conclusion of the eddying capability of the model having little impact on an AMOC response to buoyancy forcing in the SPNA might be premature and based on simulations with still insufficient eddy presence.

     Lastly, we argue that systematic model configuration comparisons like the one presented here are essential to understand the

influence of different model components and parameterizations. Our study shows that such systematic comparisons are difficult to carry out, however. Coupled and forced models naturally have different mean states. Should model parameters be optimized for each configuration or rather kept unchanged for a fair comparison? Grid resolution of each model component plays a role, too. Which resolution is sufficient for completely abandoning a related parameterization, such as GM for eddies? The presented results can certainly qualitatively support the assessment of model projections on AMOC weakening in a warming climate and

to some degree also provide quantitative guidance though each model has its own sensitivity.

## 5   Summary and conclusions

A systematic set of freshwater-release experiments with both, coupled climate and forced ocean-only model configurations using eddy parameterization and explicit simulation of mesoscale features was carried out for understanding the role of atmospheric feedbacks and mesoscale eddies in the ocean's response to enhanced Greenland runoff. We find an interactive

atmosphere to play a major role in stabilizing the AMOC against an overly strong decline in response to a multi-decadal increase in Greenland runoff. Our simulations demonstrate that mesoscale dynamics have a major impact on the regional response patterns everywhere, from the Labrador Sea to the subtropical gyre. We thus conclude that both processes should be considered for projections of regional North Atlantic changes in a warming climate.





The decline of the AMOC in the forced experiments is more than twice the magnitude of the response in the coupled
ones. This agrees well with the one forced simulation in Swingedouw et al. (2013) presenting with the strongest weakening
among a set of six, otherwise coupled models. We attribute this sensitivity to the dominance of the positive salinity feedback
in the absence of a compensating temperature feedback when the atmosphere is not able to adjust to changes in SST (Griffies
et al., 2009). Our results show up to an order of magnitude difference in the change in surface heat flux between the two
configurations. This lends strong support to the understanding that the AMOC in ocean-only simulations is too sensitive to its
own changes once triggered by external forcing, such as enhanced freshwater input.

Although having a minor impact on the AMOC response—a large-scale integrated quantity—mesoscale dynamics play a
major role in the regional distribution of and response to the freshwater added. This is due to improved representations of the
North Atlantic Current (NAC), the boundary currents and overflows in the strongly eddying simulations. The effect of eddies
in cross-frontal transport, i.e. "leaking" freshwater from the boundary current into the Labrador Sea, and restratification can be
parameterized comparatively well in non-eddying simulations using for instance the GM method (Gent and Mcwilliams, 1990).
Comparing our results with those of Böning et al. (2016), we find that meltwater concentrations in the central Labrador Sea
as simulated with a 1/20° strongly eddying model are better matched by our 1/2° simulation applying GM than the 1/4° one
without GM used as reference by Böning et al. (2016). Export and recirculation of the freshwater with the subpolar gyre
deviates significantly between non-eddying and strongly eddying models. Here, our non-eddying simulation cannot capture
sufficiently the westward propagation along the North American coast and rather simulates a massive eastward spreading near
Flemish Cap (Fig. 13). Further, the eddy parameterization fails in our case to achieve the same magnitude of exchange between
subpolar and subtropical gyres as simulated by the strongly eddying model. Obviously, the path and role of the NAC as a
strongly eddying current is less easily corrected by parameterization (Drews et al., 2015; Park et al., 2016).

We note, that model sensitivity to a freshwater perturbation is further influenced by local processes, such as overflow dy-
namics, deep mixing and wind conditions over the Greenland shelf (Ekman transport, tip jets). While atmospheric coupling is
important for the large-scale response, coupled models with coarse atmosphere grid resolution, such as used here, may suffer
from local wind biases, in particular in coastal regions. Unfortunately, the GrIS meltwater unfolds its impact involving pro-
cesses such as ocean eddies and Ekman transport from strong wind events along the narrow continental shelf of Greenland.
Thus, for a most realistic redistribution of Greenland freshwater in a model ocean sufficient resolution in both, ocean and
atmosphere, are required.

Model experiments aiming at longer, such as centennial or millennial time scales, may perform well without resolving such
processes, in particular when being used for studying responses on much broader basin-wide spatial scales. In this case, local
processes can also be skipped by the concept of the hosing experiments prescribing the freshening to the entire subpolar region.
We suspect the model mean state and internal variability to dominate sensitivity and response in this case, and local mixing
processes become less important. Also for the results presented here, we cannot rule out an influence by the mean state of
the model configuration, i.e. the initialization and the spinup length. A priori this will have a greater impact the smaller the
freshwater perturbation, which in our case is only 0.05 Sv. For such moderate perturbation Martin et al. (2022) already noted





a significant influence by internal variability of the climate system. Recent increases in freshwater flux off Greenland are even five times smaller.

Some of this may sound trivial but it cannot be overly emphasized that the subpolar North Atlantic is a highly complex ocean region and different models (or model configurations) may yield different responses to the same freshwater perturbation for different reasons. We conclude, that to seriously project the impact of enhanced Greenland runoff over the next decades a high-resolution, near Rossby radius resolving but at least 1/20° model is required. This is to resolve eddy processes important for restratification in the Labrador and Irminger Seas as well as dynamic processes at the Northwest Corner. Alternatively,

scale-aware parameterizations for such processes need to be applied to and further developed for eddying but not-quite Rossby radius resolving ocean models. In this we agree with the assessment of Hewitt et al. (2020). For projections beyond 5–10 years into the perturbed state we see a necessity to include full ocean-atmosphere interaction to account for the compensating temperature feedback of a weakening AMOC. Ocean hindcast simulations being forced with atmosphere reanalysis include this feedback to some extent though as long as the reanalysis considered observed SST and realistic, time-varying estimates of

freshwater input from Greenland are applied as boundary condition.

Despite its rapid increase over the past two decades, Greenland meltwater runoff and solid ice discharge still are relatively minor player in the freshwater budget of the subpolar North Atlantic being influenced by Arctic export, salinity variations in the Gulf Stream, precipitation and sea-ice melt. The AMOC will weaken under global warming—even without ice-sheet melt (Weijer et al., 2020). A weaker AMOC is likely less sensitive to Greenland runoff (Swingedouw et al., 2015). Climate model

experiments disagree on the potential impact of Greenland meltwater among all other consequences of global warming (e.g. Swingedouw et al., 2006; Mikolajewicz et al., 2007). And ocean models present with diverse sensitivity to freshwater added from Greenland as discussed in the present study. Therefore, it remains a major challenge to answer the question whether Greenland meltwater can and when it will potentially tip the scale on deep convection.

*Code and data availability.* FOCI1 is composed of several components, which prohibit distributing the full source code due to licensing is-

sues; ECHAM6.3 is provided by the MPI-M (https://mpimet.mpg.de/en/science/models/mpi-esm/echam); NEMO3.6 (rev. 6721) is available at https://forge.ipsl.jussieu.fr/nemo/svn/NEMO/releases/release-3.6/NEMOGCM; FOCI-specific code changes and runtime environment are provided at http://doi.org/10.5281/zenodo.3568061 (see Matthes et al., 2020). Model output from all experiments and the Jupyter notebooks required to re-produce the analysis and figures are available through GEOMAR at https://hdl.handle.net/...

## Appendix A: Selecting a reference period

Internal variability of the climate system has the potential to mask the response of the ocean to a moderate freshwater perturbation of 0.05 Sv as demonstrated by Martin et al. (2022). For the present study we have carefully chosen the time periods for computing the reference mean state and the perturbed state. We deliberately chose different approaches and periods for the coupled and forced experiments as summarized at the beginning of Section 3. Here, we present the response in AMOC strength (Fig. A1) and in SST (Fig. A2) to support our approach.



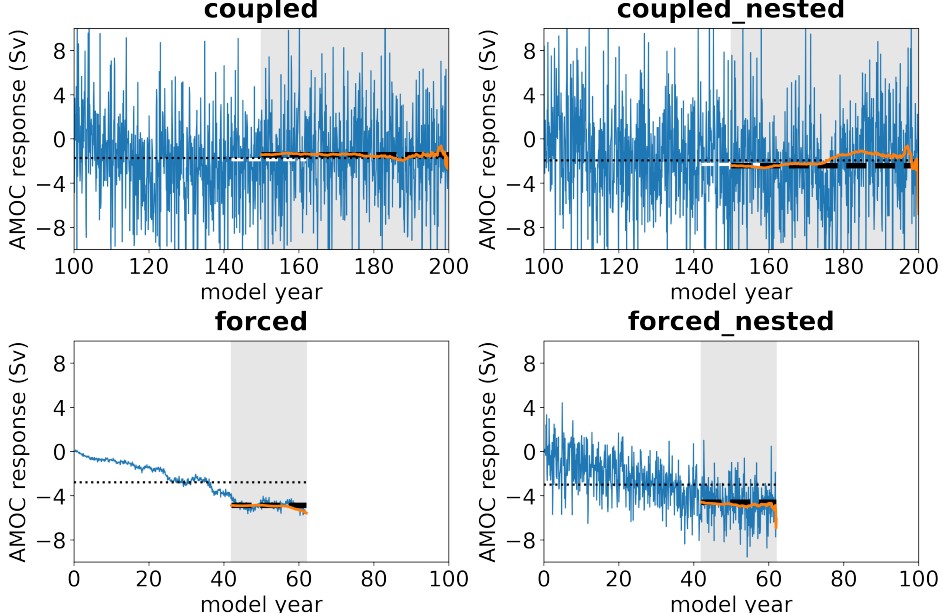

**Figure A1.** Difference in monthly mean AMOC strength (blue) between the perturbation experiment and reference run of all four model configurations. Gray shading of the background marks the respective averaging periods for computing the response to the freshwater perturbation, years 151–200 for the coupled and 43–62 for the forced experiments. The orange lines depicts the running mean based on the cumulative sum computed backwards in time starting with the last month of each simulation. Black dashed and dotted lines indicate the overall mean for the response period (gray shading) and for the entire time series, respectively. The white dashed line in the upper panels depicts the mean of model years 143–162, which corresponds to the 20-year response period of the forced experiments (cf. lower panels).

Figure A1 offers a very clear impression of the internal variability simulated by the four model configurations. While the difference in AMOC strength between the perturbation experiment and the reference run is dominated by the weakening trend of the AMOC in the forced experiment, monthly and inter-annual variability is significantly larger in the forced-nested configuration. Since atmospheric varibaility is prescribed in these simulations, we can attribute the larger variability to the explicit simulation of mesoscale eddies. Despite this enhanced variability, we can still see the multi-decadal decline at the

beginning of the experiment and that the trend in the two forced experiments is similar. In contrast, internal variability on monthly to decadal time scales dominates the time series of the two coupled configurations (Fig. A1 upper panels) and the decline at the beginning is difficult to identify. The former is also indicated by the standard deviations given in Table 2.

    This behavior of internal variability motivated our decision to compute the response to the freshwater perturbation in two different ways for the coupled and forced experiments: By construction the forced experiments experience the same temporal

evolution of atmosphere driven variability and we can directly subtract the results of the perturbed experiment from the reference run for each time slice. The white noise added by the freely running atmosphere in the coupled experiments requires a statistical approach and we compute mean state and response over longer time periods (cf. 2).





In the forced experiments we can identify the AMOC strength to reach a seemingly stabel state for the last 20 years. This is supported by computing a running mean of the AMOC response based on the cumulative sum expanding backwards in time from the last instance of the time series. This mean stays stable until approximately year 40 (orange line in Fig. A1. Similarly, we find a relatively stable state for the last 30+ years for the coupled experiments. As noted by Martin et al. (2022), the AMOC decline in the coupled experiments is difficult to seperate from internal decadal variability but the adjustment period is likely shorter than in the forced experiments due to the overall weaker response. Therefore, we simply use the second half of these experiments to improve statistics.

The AMOC strength is an integrative quantity and internal variability is typically even larger at the surface. We thus also present the influence of the averaging periods on the SST response in Figure A2. Here, we distinguish the average over the entire simulation length (long-term mean) and averages over 20 and 50 years, i.e. over model years 43–62 (forced and coupled runs) and 151–200 (coupled runs only) as depicted in Figure 2. Firstly, we note that the years 42–63 are slightly warmer compared to the respective long-term mean in the forced configurations. Secondly, 20 and 50-year reference periods differ in their spatial distribution of warm and cold anomalies with respect to the long-term mean in the SPNA of the coupled configurations. We consider these variations, which are mostly $<0.2°$C and nowhere exceed $0.6°$C , as uncertainty related to internal variability, i.e. the selection of the reference period. An exception is the Nordic Seas where sea ice retreat drives larger warming in the forced experiments.

The four lower rows in Figure A2 show differences in SST between freshwater perturbation experiments and the reference runs. Using the forced experiments as an example, we see that the response in SST is biased low when using the long-term mean in both, reference and perturbation run (EXPltm - REFltm). In contrast, warming in the Nordic Seas (cooling in the SPNA) is overestimated (underestimated) when comparing the last 20 years of the perturbation experiment with the long-term mean of the reference run (EXP20y - REFltm)). Hence, the response to the freshwater perturbation is best isolated when subtract the average of the exact same time period (EXP20y - REF20y)—at least in experiments where atmospheric forcing is prescribed. Since atmospheric variability cannot be controlled in coupled simulations and thus varies in each run, the optimal approach is to average over longer time periods to reduce the impact of internal variability but consider an adjustment period. The latter is typically shorter for surface properties than the deep ocean and for regions closer to the perturbation site, such as the SPNA in the present study, than more distant ones as shown in Martin et al. (2022).

*Author contributions.* TM and AB together conceived the idea for the present study and wrote the text. TM conducted all of the numerical experiments and carried out the analysis including production of the figures.

*Competing interests.* There are no competing interests.



*Acknowledgements.* The authors thank Franziska Schwarzkopf, Jan Harlaß, Sebastian Wahl and the FOCI development team for their invaluable support in setting up the model simulations. This study was funded by the German Federal Ministry of Education and Research (BMBF) as a Research for Sustainability initiative (FONA) through the project PalMod: From the Last Interglacial to the Anthropocene –

Modeling a Complete Glacial Cycle; WP1.2.2: Scale Interactions: Analysis of ocean dynamics (FKZ: 01LP1503D). The work was supported by the North-German Supercomputing Alliance (HLRN) providing computational resources and technical support.



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



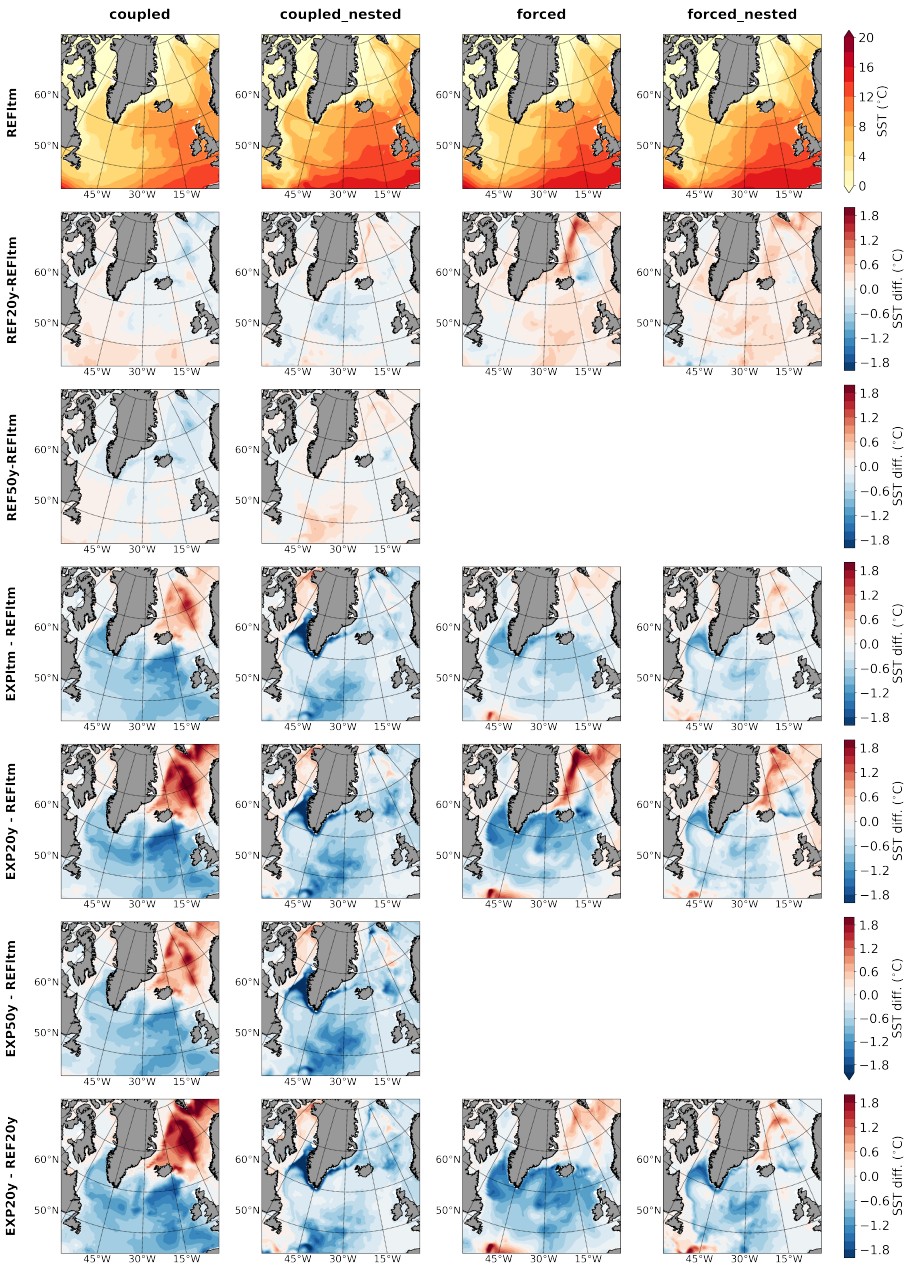

**Figure A2.** Reference mean state of sea surface temperature (SST) from all four model configurations (top row) and deviations thereof depending on either time period or freshwater perturbation or both. The labelling on the left reads as follow: REF refers to the unperturbed reference experiments, EXP to the freshwater perturbation experiments; temporal averaging is applied to either the entire run yielding a long-term mean (ltm, 100 years for the two coupled, 62 years for the two forced configurations), or to the years 43–62 (20y) and 151–200 (50y, coupled only) after onset of the perturbation.