# Peer review of "On the ocean's response to enhanced Greenland runoff in model experiments: relevance of mesoscale dynamics and atmospheric coupling"

_EGUsphere, 2022_

## Referee Comment (RC1)

**Review on : "On the ocean's response to enhanced Greenland runoff in model experiments: relevance of mesoscale dynamics and atmospheric coupling"**

Torge Martin and Arne Biastoch
Correspondence: Torge Martin (tomartin@geomar.de)

**General comment:**

I thank the authors for the submitting this manuscript, they present here a new ensemble of freshwater forcing (FWF) experiments around Greenland with four simulations each in a different models configuration. The paper reads well and show a great knowledge of the processes governing North Atlantic ocean's dynamics. Figures are clear and response of the increased freshwater is nicely described as well the role of atmospheric feedbacks and mesoscale dynamics. Summary and conclusions were particularly well-written. That being said, the paper is quite long and it would benefit from being synthesized. I also propose below some specific and technical comments in order to improve the paper.

**Specific comments:**

**1 The forcing:**
It unclear to me why the forced simulation were prescribed an atmosphere with a transient forcing while the coupled one have a preindustrial one. Comparing an historical forced simulation to a preindustial coupled simulation mixes the role of the atmospheric feedback and anthropogenic warming, why constant forcing was not used for the forced configuration?

**2 The AMOC:**
The AMOC has a large decadal to multidecadal to multicentennial variability (Ortega, 2015) depending on the model. The long term period chosen (100 years) is thus a rather short, AMOC could be experiencing a trend (see DOI: 10.1175/ JCLI-D-13-00651.1, their figure 4). By comparing the period 51-100 years to the mean state 1-100 in the coupled simulations, you are mixing the response of the FWF and internal variability. Same period would appear to be a more clear comparison rather than artificially increasing the signal by changing the period. Additionally, AMOC could not respond to FWF the same way if it is on its stronger or weaker phase, so it would be useful to have a figure showing the time series of the annual AMOC, and what was its state when the perturbation was added. Last, the mean response stays within the limits of the internal variability, so it should be mentioned that it is not significant.

**3 The periods of comparison and internal variability:**
The choice of the periods of comparison is key to this study because it impacts all the results. This question is discussed only in the appendix while it seemed rather central to me, and it could benefit for being a bit more structured and clarify. Figure A2 shows averages over periods, but we are lacking some times series to give us an idea of the decadal variability in the Labrador sea for instance. The response in the coupled simulations could be a result of the internal variability: as the system is chaotic, changes in initial conditions could lead to another state. Seems to me that taking a long-term time mean is not enough to take out the several feedback effects from the system (Swingedouw, 2007b). I understand that this paper have chosen to do one member (run) per configuration but please maybe add a paragraph discussing and clarifying this issue (forced signal from FW versus forced signal from GW etc…).

**Technical corrections:**

**1-Introduction**
l.20: add a citation after "decay"-
    "as well as" → "as well as is"
l. 58: "to shedding" → "to shed"
l.60: description of (a), (b) and (c) experiments is not clear, please describe the whole experiment in one time, for example: you will compare 4 simulations with freshwater forcing (FWF) to the same simulations without FWF and those 4 simulations are : one coupled, one forced, both with and without nest.
l.67: introduction the question of the mean state question is a bit abrupt, maybe explain a bit before line 66 why it is has to be addressed with one citation
l. 71: "by" → "from"

**2-Model configurations and experiment**
Table 1: the term "spatially varying" is bit misleading: the spatial distribution is kept constant in time right (*cf "The perturbation is constructed from the monthly-mean runoff plus discharge fluxes of Bamber et al. (2018) by averaging the period 1992–2016", line 129*)? Please clarify that either in the legend of the Table 1 or in the text
l.107: specify or give a bit more information about what "model parameter" you are referring to
l.117: "hight" → "height"
l.127: "most" → "mostly"
l. 134 "62 and 100" → missing the word "years"
l. 138: not clear why "maximum runoff in June to August" is simplification, I guess this relates to the line l.144: "shifting the seasonal peak" maybe reformulate to make it easy to follow and specify what should be the real maximum month for the runoff to get out of the fjord into the open sea
l. 148: is the error calculated here computed from he loss of tracer concentration along the experiment? Please specify

**3-Results**
l.154 and 155: "variations in internal variability" → "internal variability"
l.155: "which evolve freely within the preindustrial boundary conditions provided" → "which atmosphere evolves freely under preindustrial forcing" the term " boundary conditions" is used for regional modelling, when we prescribe values at the spatial boundaries of the model domain.  For climate simulation, better to use the term "forcing".
l. 157: suppress: "which are the same for each simulation", already said line 98 and in Table 1
l.159: "can only be expected to exist after several decades" justify this choice of time frame, maybe by adding a citation

**3.1 Ocean mean states and responses**

**AMOC**
Table 2: "Denmark Strait (DS) overflow potential density" → is it the annual mean?
Figure 4: The caption is unclear, please explain what are the dark blue histogram and  maybe add "(light blue)" after "perturbed states".
l.180: you are not coupling to the same atmosphere, the slower AMOC in the coupled simulations could be the results of the transient forcing

**Large-scale upper ocean salinity and freshening**
l.199: add reference to Figure 1 to show transportation of FW
l. 201: Salinity is decreased a lot along the western coast of Europe in the coupled non-nested simulation. Are the FW leaking towards the subtropical gyre as seen in other hosing experiment (Swingedouw, et al. 2013; Devilliers et al , 2021), maybe showing a larger map could answer that?
l. 203: I do not see a more realistic Gulf Stream separation in the coupled nested response than in the coupled response (Fig5 b, left) Please correct the statement.
l.210: 1 → 1 psu
l.219: Add a figure of the sea-ice, or "(not shown)"
l. 229 ENA is defined later (line 247)
l. 230: "Nordic Sea" → "Nordic Seas"
l. 237: "The two nested experiments both feature an overall stronger inflow of Atlantic water into the Nordic Seas" I do not see that in the figure, please explain

**Water-mass transformation:**
This subsection is 6 pages it self, far larger than AMOC, salinity and temperature responses (1 to 2 pages each), please consider to reduce it or making it a 3.2 section to have some equilibrium
l. 243: "sights" → "sites"
l. 251: which is due to a weaker AMOC in the non-eddying simulations
l. 257: but coupled simulations also present with a stronger AMOC, bringing more warm water into SPNA
l. 264: question ?
l. 269: "source waters" → "water sources"
Figure 8: add the Labrador sea shelf region to be coherent with Figure 6
l. 270: Figure 9 shows that density seems more different between Labrador and Irminger sea in forced non-nested than in coupled nested
l. 273: "Moreover, the coupled runs exhibit a stronger salinity": I see that only for the coupled nested simulations
l. 273: "thus density gradient" →  "thus stronger density gradient"
l. 275: "more detail" → "more details"
l. 277: "The freshwater perturbation leads to a freshening and cooling in the ENA and on the ENA shelf in all configurations" → I disagree: Fig 9 shows a warming in the ENA shelf for the non nested simulations and in the ENA for the forced-nested (comparing circle and cross)
l. 291: "similar pattern" →"similar pattern to ENA"

l. 295: it is consistent with the reduction of the convection activity in the Lab. Sea (Fig 7, b)

l.305: "a consequence of the shallower deep convection in the forced configuration" → "a consequence of the shallower mixed layer in the forced configuration (see Fig. 7)"

l. 318: "but the 1/10∘ ones without though" → reformulate

l. 326: "not shown", isn't it shown in Figure 5 a)?

l. 330: "(not shwon)" → "(not shown)"

l. 331: "the least" → "less". Figure 13 is cited before Figure 12, please exchange figure numbers.

l. 341: Figure 7 a) does not show that "mixing across the SPNA [...] is enhanced compared to the non-eddying configurations", not for the coupled one at least

l. 342: "in both experiments" : which ones? It decreases more in the forced than in the forced-nested

**3.2 Mesoscale dynamics**

l.348: "This is" → "These are"

l. 352: you would need the same figure at ½ degree to compare to use the word "improves", please change to "display a realistic..."

l. 354: "For example": This is not an example of why "the finer resolution [...] is inadequate to simulate the full dynamical mesoscale spectrum." please re-organize

l .356: "much closer to observations", a citation is need here to justify the numbers

l. 360-361: "over/underestimation" is not the best term since there is no comparison to observation here so we do not know if the deep mixing is over/underestimated maybe use "stronger/weaker" instead?

l.362: "in the nested perturbation experiments", please add the depth you are referring to (50 meters I guess)

l. 363: "highlight the necessity of using at least 1/20 ∘ grid resolution" → "suggest that the resolution may not be high enough with this model"

l. 370-373: add references to figures.

l. 390: "ocean below 1000 m." → "ocean below 1000 m for the configurations with eddy parameterization."

l. 395: "stronger meridional density gradient in the NAC region", add a reference to Figure

**3.3 Atmospheric coupling**

Figure 14: there is one extra parenthesis in the caption. Seems like the coupled-nested displays values on a coarser grid than the forced-nested?

l. 417: "In the non-eddying configurations," → "In the coupled non-eddying configuration,"

l. 421: "can adjust to changing" → "can adjust"

l. 426: "the upper ocean cooling [...] reduces the temperature difference between ocean and […] atmosphere" → you mean that in the forced model, it is the upper ocean who adjusts to the atmosphere to reach equilibrium? Maybe add a little more details about the surface heat flux estimation in a forced model, or a citation where this is explained

l. 455: "this results": you should should mention you are referring to the response to to FWF

l. 457: "southward expansion of the sea-ice edge" : the extension is not very clear and wind response has a lot of noise, maybe worth to be mentioned

l. 458: "The particular reinforcement of the onshore Ekman transport" is that a stable feature in the the coupled-nested configuration or period dependent? Is it more or less constant along the simulation, have you tried different time-slices?

**4 Discussion**

l.475: "decade" → "decades"

l. 487: "to hindering" → "to hinder"

l. 488: "Potentially in consequence thereof, enhanced deep convection in the Irminger Sea has certainly offset any impact of recently enhanced runoff from Greenland on deep water formation." I do not understand that statement as enhanced deep convection means impact on deep water formation

l. 493: why the plan of the results is not kept here? As: first mesoscale eddies and second atmospheric coupling

l. 503: "support stronger deep convection" you mean "support stronger reduction of deep convection"

l. 504: "surface heat loss is less than 10%" this is because atmosphere is adjusting along the simulation, not so sure this questions the importance of a positive feedback

l. 506: "to doubting" → "to doubt"

l. 510: "In" → "in"

l. 536: "The eddies resolved in our model are obviously not sufficient for bringing enough meltwater to the deep convention sites to achieve results comparable to Böning et al. (2016)." this is contradictory to the statement of before

l. 534: "larger eddies, […] which carry relatively fresh water from the boundary current into the interior Labrador Sea", so the resolution is sufficient to carry the freshwater

l. 548: "a apply" → "apply"

**5 Summary and Conclusion**

l. 599: "deviates" → "deviate"
l. 604: "We note, that" → "We note that"
l. 610: "are" → "is"

**Appendix:**

Figure A1: These figure are hard to read, please zoom in the forced and forced-nested  and add a figure showing the annual five year running mean to display the phase of the AMOC. Explanation of the orange line is unclear, maybe add a formula
l. 653: "varibaility" → "variability". "we can attribute the larger variability to the explicit simulation of mesoscale eddies" → explain a little bit more maybe how the parametrization of the mesoscale processes leads to such a lower seasonal variability
l. 659: "By" → "by"
l. 662: "(cf. 2)" → "(cf. Figure A2 or section? 2)"
l. 663: "stabel" : stable – no I rather see steady decline, since each month AMOC_perturb – AMOC_control < 0
l. 664: again, I do not understand what you are summing here
l. 665: missing parenthesis
l. 666: "30+" → "30". "we find a relatively stable state for the last 30+ years for the coupled experiments" you mean a stable difference?
l.667: "the adjustment period is likely shorter than in the forced experiments due to the overall weaker response." I am not so sure about that see general comments
l. 668: "Therefore, we simply use the second half of these experiments to improve statistics." not comparing with the same period, you are mixing the signals, maybe add a figure comparing the same period to show the difference
l. 669: "to improve statistics." : to which statistics are you referring to?
Figure A2: move it before bibliography

---

## Author Response (AR1)

**Review on : "On the ocean's response to enhanced Greenland runoff in model experiments: relevance of mesoscale dynamics and atmospheric coupling"**

Torge Martin and Arne Biastoch
Correspondence: Torge Martin (tomartin@geomar.de)

**General comment:**

I thank the authors for the submitting this manuscript, they present here a new ensemble of freshwater forcing (FWF) experiments around Greenland with four simulations each in a different models configuration. The paper reads well and show a great knowledge of the processes governing North Atlantic ocean's dynamics. Figures are clear and response of the increased freshwater is nicely described as well the role of atmospheric feedbacks and mesoscale dynamics. Summary and conclusions were particularly well-written. That being said, the paper is quite long and it would benefit from being synthesized. I also propose below some specific and technical comments in order to improve the paper.

We thank the reviewer for their overall positive feedback on our manuscript and the detailed comments, which certainly help to improve the paper. The specific comments on forcing, AMOC, internal variability and overall length of the paper are well taken and we certainly will consider all of them for the revised version.

**Specific comments:**

**1 The forcing:**

It unclear to me why the forced simulation were prescribed an atmosphere with a transient forcing while the coupled one have a preindustrial one. Comparing an historical forced simulation to a preindustial coupled simulation mixes the role of the atmospheric feedback and anthropogenic warming, why constant forcing was not used for the forced configuration?

The climate forcing indeed differs and we were weighing our options to match various project goals. The coupled, non-nested model simulations were first conducted for the multi-model comparison presented in Martin et al. (2022) and the coupled-nested simulations were executed to match these, i.e. running pre-industrial control runs to (a) clearly identify model-specific mean state and internal variability and (b) isolate the effect of Greenland meltwater input. This is standard procedure for coupled sensitivity experiments.

Another goal was to compare such experiments with typical ocean-only hindcast simulations, such as in Böning et al. (2016). While repeat-year forcing would have appeared to enable straight comparison with the pre-industrial control runs of the coupled model, we think that the historical forced simulations are advantageous for two reasons: (1) they enable a more direct comparison to the study of Böning et al. (2016) and (2) include realistic interannual to decadal atmospheric variability which would be excluded by repeat-year forcing and internal climate variability would appear largely damped. Further, we note that even by repeating the coupled experiments under historical greenhouse gas forcing would not necessarily yield a better comparability with the forced ocean-only runs because timing of global warming effects may not be exactly the same in coupled experiments.

Since there seems not to be an ideal solution, we considered internal variability the more important factor and thus ran the ocean-only experiments with variable forcing but wanted the coupled experiments comparable with the multi-model comparison presented in Martin et al. (2022, GRL).

**2 The AMOC:**

The AMOC has a large decadal to multidecadal to multicentennial variability (Ortega, 2015) depending on the model. The long term period chosen (100 years) is thus a rather short, AMOC could be experiencing a

trend (see DOI: 10.1175/ JCLI-D-13-00651.1, their figure 4). By comparing the period 51-100 years to the mean state 1-100 in the coupled simulations, you are mixing the response of the FWF and internal variability. Same period would appear to be a more clear comparison rather than artificially increasing the signal by changing the period. Additionally, AMOC could not respond to FWF the same way if it is on its stronger or weaker phase, so it would be useful to have a figure showing the time series of the annual AMOC, and what was its state when the perturbation was added. Last, the mean response stays within the limits of the internal variability, so it should be mentioned that it is not significant.

We agree that AMOC internal variability exists on multi-centennial time scales, of course. Computing trends for various subsections of the reference runs, we found weak AMOC trends for the two coupled reference runs for periods of 100 to 200 years (coupled: 0.025-0.07 Sv/decade, coupled-nested: -0.05 to +0.03 Sv/dec) but a considerably wider distribution when using periods shorten than 80 years. We thus consider a 100-year long reference period as sufficient to limit the imprint of internal variability. Regarding the AMOC strength at the onset of the freshwater experiment, the coupled-nested run starts close to the long-term mean AMOC strength (+0.5 Sv) whereas the coupled experiment presented here starts in a low phase (-2.2 Sv). However, the coupled run is part of a small ensemble discussed in Martin et al. (2022), who show a robust AMOC weakening over the last 50 years independent of the starting condition. See also response and figure further below.

**3 The periods of comparison and internal variability:**

The choice of the periods of comparison is key to this study because it impacts all the results. This question is discussed only in the appendix while it seemed rather central to me, and it could benefit for being a bit more structured and clarify. Figure A2 shows averages over periods, but we are lacking some times series to give us an idea of the decadal variability in the Labrador sea for instance. The response in the coupled simulations could be a result of the internal variability: as the system is chaotic, changes in initial conditions could lead to another state. Seems to me that taking a long-term time mean is not enough to take out the several feedback effects from the system (Swingedouw, 2007b). I understand that this paper have chosen to do one member (run) per configuration but please maybe add a paragraph discussing and clarifying this issue (forced signal from FW versus forced signal from GW etc...).

The Appendix is intended to collectively address the issue of distinguishing the response to the freshwater from internal variability and we appreciate the additional suggestions to discuss this in a more comprehensive and convincing way in the revised version. For the coupled experiments we chose 50-year means to exclude influence by internal variability as best as possible considering the overall experiment length of 100 years and a multi-decadal adjustment phase. Martin et al. (2022) show that for most of the subpolar North Atlantic a quasi-equilibrium of the response can be assumed after a few decades, such that an average over years 50-100 seems like a reasonable approach. Further, three ensemble members of the coupled configuration are discussed in that paper showing that despite internal variability interfering with the response signal the changes discussed in the present manuscript and also their magnitudes are robust (for the same configuration). We assume the same holds for the additional configurations discussed here. Because of the already extensive main part of the paper, we remain with our decision to focus this discussion in an appendix. However, we added a dedicated hint towards the appendix at the end of the introduction section.

**Technical corrections:**

**1-Introduction**

l.20: add a citation after "decay"- "as well as" → "as well as is"
l. 58: "to shedding" → "to shed"
done

l.60: description of (a), (b) and (c) experiments is not clear, please describe the whole experiment in one time, for example: you will compare 4 simulations with freshwater forcing (FWF) to the same simulations without FWF and those 4 simulations are : one coupled, one forced, both with and without nest.
good point; done

l.67: introduction the question of the mean state question is a bit abrupt, maybe explain a bit before line 66 why it is has to be addressed with one citation
Done. We address the issue of the model mean state citing Stouffer et al (2006) and Swingedouw et al. (2013) as examples in the previous paragraph now: "While earlier studies suggest no systematic dependency of the AMOC response on its reference mean strength \citep{Stoffer2006, Swingedouw2013}, we do consider a potential sensitivity to the general ocean mean state."

l. 71: "by" → "from"
done

**2-Model configurations and experiment**

Table 1: the term "spatially varying" is bit misleading: the spatial distribution is kept constant in time right (*cf "The perturbation is constructed from the monthly-mean runoff plus discharge fluxes of Bamber et al. (2018) by averaging the period 1992–2016", line 129*)? Please clarify that either in the legend of the Table 1 or in the text
The table caption now reads: "A freshwater flux (FWF) of 0.05 Sv is added as seasonally varying runoff using a spatially heterogeneous but time-invariant pattern along Greenland's coasts in the perturbation experiments."

l.107: specify or give a bit more information about what "model parameter" you are referring to
Our statement refers to diffusion and viscosity parameters set in the namelist, which typically scale with both grid resolution and time step, such as rn_ahtbbl: base/RHO^2, rn_aeiv_0 set to 0 in nest, rn_aht_0: base/RHO, rn_aht_m: base/RHO, rn_ahm_0_blp: base/RHO^2, rn_ahm_m_blp: base*RHOT/RHO^4, and rn_ahm_m_lap: base*RHOT/RHO^2 where RHO and RHOT are the spatial and temporal refinement factors.
We add "…, which mostly affects viscosity and diffusion settings." to the respective sentence.

l.117: "hight" → "height"
corrected

l.127: "most" → "mostly"
This change would give the sentence a different meaning. We decided to rather remove "most".

l. 134 "62 and 100" → missing the word "years"
added; thanks for noting these small glitches

l. 138: not clear why "maximum runoff in June to August" is simplification, I guess this relates to the line l.144: "shifting the seasonal peak" maybe reformulate to make it easy to follow and specify what should be the real maximum month for the runoff to get out of the fjord into the open sea
The respective paragraph begins with listing three simplifications of which the seasonal timing is one. In the reminder of the paragraph we provide further detail ("meltwater runoff is first entrained into the fjord circulation, which causes both a vertical redistribution and a temporal delay of several weeks before entering the open ocean") and arguments for the feasibility of the simplification ("prescribed freshwater … also shifting the seasonal peak by a month."). We consider this sufficient detail but note here, that the Bamber et al. data set has peak runoff in June to August, which is why it is a simplification to simply prescribe this runoff field without further treatment of the seasonal timing.

The delay between runoff (and calving) into the fjord and meltwater being exported into the open ocean varies depending on fjord circulation and topography. It is thus not possible to provide a "real maximum month".

l. 148: is the error calculated here computed from he loss of tracer concentration along the experiment? Please specify
The sentence now begins with "Using the passive tracer concentrations, we compute an error …"

**3-Results**

l.154 and 155: "variations in internal variability" → "internal variability"
"variations in" removed

l.155: "which evolve freely within the preindustrial boundary conditions provided" → "which atmosphere evolves freely under preindustrial forcing" the term " boundary conditions" is used for regional modelling, when we prescribe values at the spatial boundaries of the model domain. For climate simulation, better to use the term "forcing".
agreed and corrected

l. 157: suppress: "which are the same for each simulation", already said line 98 and in Table 1
done

l.159: "can only be expected to exist after several decades" justify this choice of time frame, maybe by adding a citation
done, citing Swingedouw et al. (2013) and Jackson and Wood (2018) for example

**3.1 Ocean mean states and responses**

**AMOC**

Table 2: "Denmark Strait (DS) overflow potential density" → is it the annual mean?
Yes, in fact also decadal running mean. We rephrased the sentence: "Correlation coefficients (Pearson's $r$) between annual-mean AMOC strength and Denmark Strait (DS) overflow potential density ($p_{ref}=0$) as well as March-mean mixed-layer depths (MLD) in the Labrador Sea after applying a decadal boxcar averaging filter to all time series"

Figure 4: The caption is unclear, please explain what are the dark blue histogram and maybe add "(light blue)" after "perturbed states".
The blue perturbed state histograms are transparent, i.e. "dark" blue shading indicates underlying black histogram. We added "(blue, transparent)" after "perturbed states".

l.180: you are not coupling to the same atmosphere, the slower AMOC in the coupled simulations could be the results of the transient forcing
true, could be. Sentence rephrased: "Furthermore, the coupled configurations simulate a stronger AMOC than their forced counterparts, which could either be related to coupling with an interactive atmosphere as in \citet{Hirschi2020} or to comparing the historical (forced) with the pre-industrial climate state (coupled)."

**Large-scale upper ocean salinity and freshening**

l.199: add reference to Figure 1 to show transportation of FW
done

l. 201: Salinity is decreased a lot along the western coast of Europe in the coupled non-nested simulation. Are the FW leaking towards the subtropical gyre as seen in other hosing experiment (Swingedouw, et al. 2013; Devilliers et al , 2021), maybe showing a larger map could answer that?

Yes, freshwater leaks into the subtropical gyre as described in the named references. For most of our manuscript we prefer to keep the focus on the subpolar North Atlantic. The exchange with the subtropical gyre in non-eddying and strongly eddying simulations is subject of another forthcoming paper. However, we have expanded Figure 13, the maps of meltwater tracer concentration, southward to include and illustrate both the eastern upper and western deep export routes.

l. 203: I do not see a more realistic Gulf Stream separation in the coupled nested response than in the coupled response (Fig5 b, left) Please correct the statement.
The improved Gulf Stream location in the nested configuration is shown in Figure 3, e.g. by the reduced warm bias at the US coast (35-40°N). The sentence addressed here describes what happens in the simulation and what is later discussed in conjunction and supported by Figure 13. We add references to Figures 1 and 2.

l.210: 1 → 1 psu
we consider salinity to be unit-less

l.219: Add a figure of the sea-ice, or "(not shown)"
reference to Figure 7 added at end of sentence referring to both sea ice and mixed layer changes

l. 229 ENA is defined later (line 247)
indeed, thanks for catching this, abbreviation now defined here

l. 230: "Nordic Sea" → "Nordic Seas"
corrected

l. 237: "The two nested experiments both feature an overall stronger inflow of Atlantic water into the Nordic Seas" I do not see that in the figure, please explain
True, this is not directly shown here, we thus add "(not shown)" to the sentence. The statement is based on the fact that the subpolar gyre circulation is generally stronger in the nested configurations. While SPG strength is not discussed in detail, we do note in subsection "Boundary currents" a stronger western boundary current for the nested setup.

**Water-mass transformation:**

This subsection is 6 pages itself, far larger than AMOC, salinity and temperature responses (1 to 2 pages each), please consider to reduce it or making it a 3.2 section to have some equilibrium
We have shortened this section along with other parts of the paper.

l. 243: "sights" → "sites"
done

l. 251: which is due to a weaker AMOC in the non-eddying simulations
No, this is due to the misrepresentation of the NAC, it's too zonal placement (c.f. Fig. 2 and 5). We added this reason for the fresh bias to the sentence.

l. 257: but coupled simulations also present with a stronger AMOC, bringing more warm water into SPNA
Good point. We considered this aspect more carefully, also in reference to Figure 2, and changed the statement according to your comment: "With 10–11°C the potential temperature is very similar for all model configurations except for the coupled, non-eddying one, in which the ENA region is strongly influenced by the cold bias with respect to late 19th century reanalysis (see \reffig{fig_bias}b). The forced experiments must thus be considered relatively cool running with historical atmospheric forcing but having a weaker AMOC and hence less northward heat transport."

l. 264: question ?
comment unclear

l. 269: "source waters" → "water sources"
We consider "source waters" correct oceanographic terminology here, since we address the source water masses of the T,S properties found in the upper Labrador and Irminger Sea.

Figure 8: add the Labrador sea shelf region to be coherent with Figure 6
Comment not quite clear: do you mean Figures 8 and 9? we tried adding a frame for the Labrador Sea shelf to Figure 8 but the plot became too crowded. Instead we add a more precise definition to the caption of Figure 9: "The shelf is defined as areas shallower than 500~m in the region 62-46˚W and 56-65˚N, i.e. within the same geographical box as the deep, interior Labrador Sea"

l. 270: Figure 9 shows that density seems more different between Labrador and Irminger sea in forced non-nested than in coupled nested
l. 273: "Moreover, the coupled runs exhibit a stronger salinity": I see that only for the coupled nested simulations
l. 273: "thus density gradient" → "thus stronger density gradient"
l. 275: "more detail" → "more details"
this paragraph has been rewritten: "In both regions the coupled reference simulations are a little saltier than their forced counterparts and the same holds for the comparison between eddying and respective non-eddying simulations. We suggest a lack of offshore Ekman transport in the coupled configurations and generally stronger deep convection in the nested ones as causes and discuss these further below. Another more obvious salinity and thus density difference is found between the boundary current (small purple circles) and the interior Labrador Sea (purple). This difference is significantly smaller in the non-eddying than in the respective eddying simulations and can be related to a insufficient exchange across the shelf break in the latter (more details in \refsec{sec_meso_dyn})."

l. 277: "The freshwater perturbation leads to a freshening and cooling in the ENA and on the ENA shelf in all configurations" → I disagree: Fig 9 shows a warming in the ENA shelf for the non nested simulations and in the ENA for the forced-nested (comparing circle and cross)
We do not agree with this statement and think there may have been a mix-up of the color coding of ENA shelf and Nordic Seas. It is true though, that there is a slight but non-significant warming on in the ENA shelf in forced_nested configuration.

l. 291: "similar pattern" →"similar pattern to ENA"
added

l. 295: it is consistent with the reduction of the convection activity in the Lab. Sea (Fig 7, b)
right, noted in text

l.305: "a consequence of the shallower deep convection in the forced configuration" → "a consequence of the shallower mixed layer in the forced configuration (see Fig. 7)"
changed accordingly

l. 318: "but the 1/10◦ ones without though" → reformulate
sentence reformulated: "As will be further discussed below this is related to running the ocean model at 1/2\textdegree grid resolution with an eddy parameterization but at 1/10\textdegree without. However, the higher resolution …"

l. 326: "not shown", isn't it shown in Figure 5 a)?
true indeed; reference to Fig. 5a added, thank you

l. 330: "(not shwon)" → "(not shown)"
done

l. 331: "the least" → "less". Figure 13 is cited before Figure 12, please exchange figure numbers.
wording changed; Figure 12 is referenced first just 3 lines above (was line 328)

l. 341: Figure 7 a) does not show that "mixing across the SPNA [...] is enhanced compared to the non-eddying configurations", not for the coupled one at least
statement removed as part of shortening the entire "Water mass transformation" section

l. 342: "in both experiments" : which ones? It decreases more in the forced than in the forced-nested
see above

**3.2 Mesoscale dynamics**

l.348: "This is" → "These are"
corrected

l. 352: you would need the same figure at 1/2 degree to compare to use the word "improves", please change to "display a realistic..."
done, see next comment

l. 354: "For example": This is not an example of why "the finer resolution [...] is inadequate to simulate the full dynamical mesoscale spectrum." please re-organize
first part of paragraph restructured: "The ocean-grid refinement yields realistic dynamics in the nest region (Fig. 1a). We find a strongly eddying ocean where the 1/10\textdegree grid sufficiently resolves the Rossby radius, which is the case south of approximately 50$^\circ$N. In higher latitudes the finer resolution yields stronger and more focused boundary currents, such as in the Nordic Seas and the Labrador as well as Irminger Sea. For example, the western boundary current transport in the Labrador Sea at 53\textdegN of the coupled model amounts to 33~Sv and that of coupled-nested to 53~Sv, which is much closer to observations. The grid refinement significantly improves mesoscale variability over large parts of the SPNA (Fig. 2a) but is inadequate to simulate the full dynamical mesoscale spectrum north of 50$^\circ$N. Nevertheless, we find individual WGC eddies-"

l .356: "much closer to observations", a citation is need here to justify the numbers
sentence rephrased and citation added: "For example, the western boundary current transport in the Labrador Sea at 53˚N (below 400~m) amounts to 19.4~Sv and 39.3~Sv in the coupled and coupled-nested configurations, respectively, where observations yield an estimate of 30.2~Sv \citep{Zantopp2017}"

l. 360-361: "over/underestimation" is not the best term since there is no comparison to observation here so we do not know if the deep mixing is over/underestimated maybe use "stronger/weaker" instead?
text rephrased accordingly

l.362: "in the nested perturbation experiments", please add the depth you are referring to (50 meters I guess)
added: "… over the entire water column but most pronounced at 50~m depth in Figure 13."

l. 363: "highlight the necessity of using at least 1/20◦ grid resolution" → "suggest that the resolution may not be high enough with this model"
done

l. 370-373: add references to figures.
added

l. 390: "ocean below 1000 m." → "ocean below 1000 m for the configurations with eddy parameterization."
added but this is "for the nested configurations."

l. 395: "stronger meridional density gradient in the NAC region", add a reference to Figure
This is somewhat visible on the salinity and temperature fields presented in Figures 5 and 6. However, the statement is rather speculative and we decided to remove the sentence.

**3.3 Atmospheric coupling**

Figure 14: there is one extra parenthesis in the caption. Seems like the coupled-nested displays values on a coarser grid than the forced-nested?
extra parenthesis removed. Yes, surface fluxes in the coupled configurations is computed on the coarser atmospheric grid. We mention this in what was originally line 414f: "The block-like structure in the SHF output of the coupled configurations is due to the surface fluxes being computed on the coarser grid of the atmospheric model at a horizontal resolution of about 1.9°"

l. 417: "In the non-eddying configurations," → "In the coupled non-eddying configuration,"
No, this holds for both non-eddying configuraitons (see Fig. 14a). In fact, remnants of this feature are visible in the long-term mean of the forced-nested configuration as well.

l. 421: "can adjust to changing" → "can adjust"
done

l. 426: "the upper ocean cooling [...] reduces the temperature difference between ocean and [...] atmosphere" → you mean that in the forced model, it is the upper ocean who adjusts to the atmosphere to reach equilibrium? Maybe add a little more details about the surface heat flux estimation in a forced model, or a citation where this is explained
No, the upper ocean cools in the perturbation experiments as a consequence of AMOC weakening. Since (1) surface heat fluxes are driven by the temperature difference between ocean and atmosphere, (2) in winter the atmosphere is colder than the ocean and (3) the SST decrease but atmospheric temperature is unchanged in the forced experiments, the decrease in SST drives a reduction in surface heat fluxes.

l. 455: "this results": you should mention you are referring to the response to FWF
done (actually by adjusting the previous sentence)

l. 457: "southward expansion of the sea-ice edge" : the extension is not very clear and wind response has a lot of noise, maybe worth to be mentioned
Since we look at 50-year averages this would need to be wind variations on almost centennial time scale. The sea-ice edge position and associated surface heat flux changes are the only systematic changes we found and they coincide with wind stress decrease over Davis Strait. We agree that the sea-ice change is rather small but so is the wind-stress change.

l. 458: "The particular reinforcement of the onshore Ekman transport" is that a stable feature in the coupled-nested configuration or period dependent? Is it more or less constant along the simulation, have you tried different time-slices?
Yes, this is a stable feature.

**4 Discussion**

l.475: "decade" → "decades"
changed to "last decade"

l. 487: "to hindering" → "to hinder"
corrected

l. 488: "Potentially in consequence thereof, enhanced deep convection in the Irminger Sea has certainly offset any impact of recently enhanced runoff from Greenland on deep water formation." I do not understand that statement as enhanced deep convection means impact on deep water formation
Enhanced runoff from Greenland is expected to reduce deep convection, first and foremost in the Labrador Sea. A coincidental increase in deep convection in the Irminger Sea could compensate for the lack of deep water formation in the Labrador Sea and hence offset the impact by enhanced Greenland runoff. The sentence was rephrased: "Recently enhanced

deep convection in the Irminger Sea \citep{Ruehs2021} may have compensated a lack of deep water formation in the Labrador Sea and hence offset an impact by recently increased runoff from Greenland."

l. 493: why the plan of the results is not kept here? As: first mesoscale eddies and second atmospheric coupling
In the present order the last presented results are revisited and discussed first.

l. 503: "support stronger deep convection" you mean "support stronger reduction of deep convection"
No, we actually mean that the deep ocean heat and salinity bias help to maintain the ongoing deep convection in the coupled model. Sentence is rephrased: "This may help to maintain the mode of recurring deep convection making the coupled ocean less susceptible to the prescribed moderate freshwater perturbation."

l. 504: "surface heat loss is less than 10%" this is because atmosphere is adjusting along the simulation, not so sure this questions the importance of a positive feedback
good point. Statement changed to "However, we cannot exclude a significant influence by the ocean and climate mean state, which differs between coupled and forced experiments."

l. 506: "to doubting" → "to doubt"
see last comment above

l. 510: "In" → "in"
but the sentence after the colon is complete by itself and thus starts with a capital letter

l. 536: "The eddies resolved in our model are obviously not sufficient for bringing enough meltwater to the deep convention sites to achieve results comparable to Böning et al. (2016)." this is contradictory to the statement of before l. 534: "larger eddies, [...] which carry relatively fresh water from the boundary current into the interior Labrador Sea", so the resolution is sufficient to carry the freshwater
The resolution is sufficient to carry some(!) freshwater by eddies into the Labrador Sea but not a sufficient amount in total. We adjust the sentence accordingly: "The WGC eddies resolved in our model are not numerous and hence not sufficient for bringing …"

l. 548: "a apply" → "apply"
done

**5 Summary and Conclusion**

l. 599: "deviates" → "deviate"
corrected

l. 604: "We note, that" → "We note that"
corrected

l. 610: "are" → "is"
corrected

**Appendix:**

Figure A1: These figure are hard to read, please zoom in the forced and forced-nested and add a figure showing the annual five year running mean to display the phase of the AMOC. Explanation of the orange line is unclear, maybe add a formula
The idea of Fig. A1 using the same y-axis scaling for all plots is to emphasize the amount of internal variability adding noise to the AMOC timeseries in all the experiments. Sources of noise are the interactive atmosphere in the coupled runs and mesoscale eddies in the nested ones.

We rephrased the explanation of the orange line: "This is supported by computing a running mean of the AMOC strength difference between perturbed and reference run using a boxcar window always anchored at the end of the time series and expanding backwards in time." The expanding running mean informs us about the number of years prior to the end of the experiment that can be included without significantly changing the mean AMOC response occurring towards the end of the run.

The paper is already extensive and includes a number of figures. We thus add here for the purpose of the discussion the requested plot of AMOC strength time series (Note, gray lines in the upper right panel for the coupled experiments depict two more ensemble members shifted by +/- 5Sv for visibility, which are not included in the present manuscript but in Martin et al., 2022). As stated above most perturbation experiments start in a phase of relatively strong AMOC but as demonstrated by the ensemble members for the coupled configuration this has no significant impact on the AMOC response.

[Figure]

l. 653: "varibaility" → "variability". "we can attribute the larger variability to the explicit simulation of mesoscale eddies" → explain a little bit more maybe how the parametrization of the mesoscale processes leads to such a lower seasonal variability

typo corrected; we add the following condensed sentence to briefly comment on the GM-parameterization: "The eddy paramterization by \citet{Gent1990} adds isopycnal mixing to non-eddying simulations, which otherwise would lack the conversion of potential to kinetic energy from local baroclinic instability, but misses additional sub-grid scale effects and kinetic backscatter, and hence rather acts to smooth variability \citet[e.g.][]{Zanna2017,Hewitt2020}."

l. 659: "By" → "by"
opposed, complete sentence after colon may start with capital letter

l. 662: "(cf. 2)" → "(cf. Figure A2 or section? 2)"
meant Fig. 2, corrected

l. 663: "stabel" : stable – no I rather see steady decline, since each month AMOC_perturb–AMOC_control<0
typo corrected; stable because difference does not grow further, we rephrase: "stable state of difference from the reference run"

l. 664: again, I do not understand what you are summing here
rephrased, see our reply for Figure A1 above

l. 665: missing parenthesis
added, thanks

l. 666: "30+" → "30". "we find a relatively stable state for the last 30+ years for the coupled experiments" you mean a stable difference?
yes, see above

l.667: "the adjustment period is likely shorter than in the forced experiments due to the overall weaker response." I am not so sure about that see general comments
The entire sentence is: "As noted by Martin et al. (2022), the AMOC decline in the coupled experiments is difficult to separate from internal decadal variability but the adjustment period is likely shorter than in the forced experiments due to the overall weaker response." We clearly acknowledge the difficulty of separating signal from noise in the coupled runs. We thus need to make an assumption for the duration of the adjustment period and consider it as shorter for a weaker AMOC decline. Martin et al. (2022) also present a timeseries of stronger AMOC decline under twice as strong freshwater perturbation, where the adjustment period is clearly longer.

l. 668: "Therefore, we simply use the second half of these experiments to improve statistics." not comparing with the same period, you are mixing the signals, maybe add a figure comparing the same period to show the difference
There is no "same period" because the experiments are too long and internal variability causes deviations between reference run and perturbation experiment over the course of the perturbation time window. All we can do is to best estimate the noise caused by the internal variability to see whether the deviation due to the perturbation is significant. Therefore, we use as extensive periods as reasonable.

l. 669: "to improve statistics." : to which statistics are you referring to?
Any mean, difference, standard deviation we use; sentence rephrased: "to reduce noise from internal variability for improved statistics."

Figure A2: move it before bibliography
caused by the latex template, needs to be done by layout later

**Comment on egusphere-2022-869**

Anonymous Referee #2

Referee comment on "On the ocean's response to enhanced Greenland runoff in model experiments: relevance of mesoscale dynamics and atmospheric coupling" by Torge Martin and Arne Biastoch, EGUsphere, https://doi.org/10.5194/egusphere-2022-869-RC2, 2022

This paper explores how the North Atlantic Ocean responds to enhanced Greenland melt, using a suite of ocean modelling experiments. The authors carefully explore this problem by using twin experiments, with and without Greenland melt, while also examining the role of resolution (through the inclusion of high resolution nests) and forcing (by considering coupled as well as forced ocean model experiments). The authors also run their experiments for a length (100 years) sufficient to allow signals and different behaviors between experiments to develop. Key results include a compensating temperature feedback in the coupled simulations, which also have greater stability. Additionally, mesoscale dynamics, represented in the nests, play a key role, including penetration of freshwater in the sub-tropics.

This is an important topic, and the study nicely examines many aspects. The paper is generally well written and clear, with high quality figures. Thus, it definitely deserves publication, with EGU Sphere being an appropriate journal. That said, there are some ways the manuscript can be improved. There are some minor wording issues (such as unneeded adjectives). The manuscript also feels long, and given that it covers so much space, there are times that it feels like the main big picture goals get lost in the many details. So it might be good to try to tighten up the manuscript and make sure the focus is always on the main ideas and results. There are also a few technical items that could use further discussion.

We appreciate the overall positive and constructive comments by the reviewer. We will provide a detailed response together with the revised manuscript and will here just briefly touch upon the main points of the criticism. In our revision we will also try and tighten the manuscript as suggested here and also by Reviewer #1 and clarify the technical issues.

Salinity Restoring: This is first mentioned at line 102-103 when the authors mention they use a weak restoring. It would be good to explain why this is included. Also, given the authors are looking at salinity signals for Greenland melt, I have concerns about those signals being damped by this term. At the very least this is worth further discussion. Some comparison with other studies that don't use restoring, or have restoring of different strengths, would be good. Ideally, and even though the experiments with the nests are computationally expensive, it would be good to see what would happen if they were run without restoring, or at least compared to a 10 year integration period with the restoring.

In ocean-only models, salinity restoring is required to stabilize the AMOC in the forced configurations. The prescribed atmospheric forcing of the ocean-only model tends to create a fresh bias in the subpolar North Atlantic, which otherwise sends the AMOC on a declining trend. The restoring of sea surface salinity (SSS) as applied here is standard procedure and applied to many ocean-only models (e.g. Danabasoglu et al., 2014). For the simulations shown here, we emphasize that we apply a relatively weak correction ("rn_deds=-33.33 mm/d, i.e. 180 days for the model's top layer"), which is added to the surface freshwater flux but not under sea ice and not at grid nodes with runoff. In addition, we limit the SSS change per time step due to restoring to 0.5 psu (Behrens et al., 2013; Danabasoglu et al., 2014). This information is added to the manuscript as follows: "… applying a weak sea-surface salinity restoring towards the PHC3.0 climatology in the open ocean ($rn\_deds=-

$33.33~mm/d, i.e. 180~days for the model's top layer, and $rn\_sssr\_ds=$0.5 additionally limits the associated salinity change) being added to the surface freshwater flux (c.f. Behrens et al., 2013; Danabasoglu et al., 2014)."

This being said, the restoring flux does indeed compensates some of the freshwater perturbation—or rather the response in P–E (net precipitation)—on basin scale.

[Figure]

*Figure*: *(upper row) Annual mean freshwater fluxes (FWF) and their decomposition integrated over the subpolar North Atlantic, incl. Nordic Seas and Baffin Bay. The total FWF includes P-E, runoff, SSS_restoring and FW perturbation; sea-ice melt is a separate contribution. All solid lines depict the reference run, dashed lines the perturbation experiment, the sea-ice melt flux barely changes, runoff and perturbation are climatological fields. (lower row) Differences of these fluxes between reference and perturbed run.*

Firstly, we note that the integrated restoring flux is always negative in the non-eddying configuration whereas it varies around zero in the nested, eddying one (solid orange lines in upper row panels in Figure above). This stronger freshwater withdrawal is necessary in the former because the simulation features a major fresh bias in the central North Atlantic (see Figure 3c in the manuscript). Secondly, the restoring flux accumulated over the SPNA (see figure above) is smaller than the freshwater perturbation injected, which holds in especially for the nested configuration (upper right panel) but less so for the non-eddying one (upper left). And thirdly, the cooling of the subpolar North Atlantic in the perturbation experiment reduces evaporation while precipitation is prescribed and hence unaffected by the simulated ocean state. Therefore, the freshwater perturbation creates an increase in the freshwater surface flux to the ocean. The restoring acts quite efficiently against this being of similar magnitude but opposite sign on basin scale (see figure above, bottom row). The restoring

thus mitigates part of the missing negative temperature feedback in the forced experiments by compensating for the temperature feedback on the surface freshwater flux (dotted black arrow T -> F in Figure 1 of Griffies et al. (2009)). The lack of atmospheric feedbacks still affects the surface heat flux acting to diminish the negative temperature feedback associated with a change in AMOC strength. All this is now also discussed in the paper (section Discussion/Atmospheric coupling).

Re-running the perturbation experiments without SSS restoring (or with restoring from reference runs prescribed) is beyond the timeline of the manuscript submission. Being able to quantify the effect of the restoring and explain its role here is sufficient, we think.

We are not aware of similar studies explicitly stating the role of SSS restoring in scenarios with freshwater perturbation ("hosing"), which could be used for a comparison. A systematic study on the influence of surface salinity restoring is found in Behrens et al. (2013), which is now referenced in the manuscript.

Historical vs Pre-Industrial: This is first discussed for lines 110-112. I know the authors work to justify this choice later in the paper, but I think this choice needs greater justification and discussion.

As we replied to Reviewer #1, we note that running the ocean-only experiment with historical (instead of repeated year) forcing but the coupled ones under pre-industrial control conditions was a compromise to have sufficient internal variability in the former and to isolate the impact of fresh water from other global warming signals in the latter. This will be further discussed in the detailed response and argued for in the revised version of the paper.

Averaging Periods: The authors explain why they use different averaging periods, and add Appendix A as a justification. This still feels like a concern in the transient experiments, since a longer averaging period means more Greenland meltwater added to the ocean, and a longer period that potentially means in can propagate farther. I would like to see some comparison with averaging over the same period, to help confirm that the results are not being biased by the variable averaging periods.

The effect of the averaging period on the response of the AMOC is, for example, included in Figure 4 and Table 2, where we also show the distribution and mean response for the 20-year period of 43-62 years after onset of the perturbation in the coupled runs. We discuss that this result is more prone to be influenced by multi-decadal variability. We argue against an expansion of the averaging period of the forced experiments to 50 years using Figure A1, which shows a clear trend in AMOC decline prior to year 40 of the simulation. The comment of having more freshwater added and allowing it to spread further by selecting a slightly later and longer averaging period for the coupled experiments is an interesting aspect. This certainly would be an issue in identifying time scales of the responses, which we refrain from doing, and focusing on the large-scale response patterns, it is again the coupled configurations with the later/longer averaging period, which show the weaker responses (despite having more freshwater added in the end). Figure A2 compares the SST response for different averaging periods (e.g. 20 and 50 years), and while there are local differences, the large-scale patterns are robust. Based on such investigations, we concluded that it is rather internal variability than the timing of the averaging period that causes the larger uncertainty and decided for a longer averaging period for the coupled experiments. Nevertheless, we will improve this discussion in the manuscript using the concerns expressed by the reviewer as guidance.

Specific points:

L23-25: "leaving the ice sheet at a negative net mass balance" doesn't read well.
rephrased to "so that the ice sheet's net mass balance was negative in each of the last 25 years"

L38: Note sure exactly what "indicate robustly" means
rephrased to "consistently show"

L71: "most critical improvements by the grid refinement" doesn't read well.
replacing "most critical" with "major"

L87: "including entire Greenland" is missing some words/explanation
rephrased: "… in the Atlantic to study subpolar processes and to include the entire coastline of Greenland for …"

L88: Might be worthwhile to clear explain what is meant by a strongly eddy ocean
adding "…, i.e. resolves the Rossby radius, …" for explanation

L92: McWilliams
done

L99: extended
preferring present tense here

L113: "much more" – much isn't needed, more pronounced says the same thing
correct

L113: What exactly is "strongly meandering" compared to just meandering?
removed "strongly"

L117: height
corrected

L122-123: "promotes intensified, partly overly pronounced deep convection" doesn't read well
rephrased to "yields more intense, sometimes overly strong deep convection"

L131: by "data extending beyond Greenland is not considered though", do you mean you haven't included the other non-Greenland glaciers in the dataset? If so, say it directly.
agreed, statement is now a separate sentence: "Data from glaciers outside of Greenland is not considered."

L131: on the annual mean
corrected

L133: What does "over 62 and 100" mean?
Figure 3 caption: What is an "Examplary improvements"?
changed to "Examples of improvements"

L142: Half of the icebergs melting in fiords – this needs to be referenced.
this would be Enderlin et al. (2016, doi:10.1002/ 2016GL070718.), citation added to manuscript

L144-145: "we find the prescribed freshwater rapidly mixed over the depth of the Greenland shelf by the ocean model also shifting the seasonal peak by a month" – no idea what this statement is trying to say.

Rephrased: "… we find that the ocean model effectively distributes the prescribed freshwater over depth on the Greenland shelf whereby a delay or accumulation arises so that the seasonal peak in freshwater content on the shelf is shifted by about a month compared to the prescribed perturbation."

L155 "much more pronounced" – more pronounced is good enough – the additional adjective doesn't really add anything
dropped "much", thanks for explaining

L169-171: Reference each of the listed process that the authors suggest the overturning is sensitive too.
We consider the dependencies between AMOC strength and subpolar North Atlantic temperature, salinity, surface heat flux and deep mixing as common knowledge. Also, this introductory sentence simply lists the parameters that we will investigate and discuss in the results section.

L175: Would be good to compare the model overturning strengths to observations, such as RAPID and OSNAP. Even if the paper's focus is understanding responses in different model configurations, it helps to understand the realism of different results/measures.
We added the following statement to provide an idea of a valid range for AMOC strength: "For comparison, the observed strength of the AMOC at 26.5˚N is 17-18 Sv with a standard deviation of 3.4 Sv in monthly data (McCarthy et al., 2015; Biastoch et al., 2021)."

L187 "much more sensitive" – more sensitive is fine. Also, given the declines in Sv, might it be worth mentioning the percentage changes?
We removed "much" (also from other occurrences of "much more" in the text) and added percentages based on values given in Table 2.

Paragraph ending on L205: How does this propagation compare with other, previous, studies of Greenland melt.
We added the following sentence at the end of the paragraph: "The described redistribution pathways agree with earlier tracer simulations by \citet{Dukhovskoy2016} and \citet{Boening2016}."

L208: 'well' not needed, expressed is fine.
removed

L211: What is meant by the 'very eastern side'?
removed "very"; this addresses Figure 5b (as noted), specifically the freshening east of about 15˚W. The sentence now reads: "… on the eastern side of the SPNA (east of $\sim$15$^{\circ}$W), especially on the European shelf, …"

L220: coupled experiments – should it be plural? I.e. Is this behavior occurring in all coupled experiments?
No, this is intentionally excluding only the coupled (but not coupled_nested) experiment. And we found in the meantime, that it is mostly reduced evaporation (due to expanding sea ice) rather than sea-ice melt. Main text was amended accordingly.

L223: "Averaged over the top 200 m representative for the upper ocean" doesn't read well.
removed "representative for the upper ocean" (see next comment)

L224: "in some areas on annual mean" doesn't read well.
the sentence was rephrased:" Averaging over the top 200~m and all seasons, this cooling can reach 2.1--2.3˚C in some areas."

L226: "This is except for" doesn't read well.
replaced by "An exception is"

L230: Nordic Seas.
corrected

L236: What is meant by "the mixed layer rather shoals"?
Replaced by "and accompanied by mixed-layer depth reduction in this case."

L238: Local regions of warming – Is this significant? Or just a minor detail?
rather minor detail, sentence removed

L243: sites
corrected

L243: "For simplification, we only show the spatial means for these areas and averaged over the top 200 m if not mentioned otherwise." Isn't clear and doesn't read well.
rephrased: "Further, we confine the volume-weighted averages of potential temperature and salinity to the top 200~m."
L244: Why does this comment about grid cell averaging suddenly appear? Is there any other way to compute averages on model grids where the area/volume spatially varies?
No, statement removed as it is the natural way to compute spatial means on grids.

L247: remove "being"
done

L255: "the large scale we focus on" – be quantitative, which will help this discussion.
replaced with "basin-scale changes we focus on"

L260: "...have approximately the same..."
done

L264-265: Why is the overflow water warmer if the mixed layers are deeper in the Nordic Seas?
Because a greater water volume is ventilated and would need to be cooled through approx.. the same surface area and over the same time period.

L280: "barely is a density change noticeable" doesn't read well.
replaced by "… the density change is minor."

L284: "presents with excessive freshening" isn't clear.
rephrased: "…where we find much greater freshening in the ENA."

Figure 10: Why use a line a latitude (60N) instead of the observational OSNAP line – would be useful for readers to look at the model fields where observations exists.
We have adjusted Figure 10 to show the model fields along the OSNAP cross-section. All main characteristics of the results also hold for this cross-section, which is located farther south on the western side of the Labrador Sea. Because of this shift, the section does not run through the main deep convection site of the models anymore and displayed mixed-layer depths are reduced in comparison to the line along 60N.

L289: Would a reference to Behrens et al be useful here?
agreed, citation added

L294: "it does not become clear" doesn't read well.
changed to "…it is not apparent from…"

Figure 11: Is MLD > 500 m appropriate for the ENA shelf? Additionally, can you try to estimate a formation rate in Sv, to help readers put the numbers in context compared to other studies? Also, how realistic are the areas of deeper MLs in the various simulations?
Deep convection in the ENA shelf region reaches as deep as 800m (in coupled configuration even 1000m) in March in the long-term mean of the reference experiment except for the forced configuration. Note, this is in fact the deep ocean region between 15°W and the European continental shelf.

L307: "enabling properties of the initialization fields still visible" – not sure what the authors are trying to say here.
sentence split and rephrased: "… coupled experiment (1500 years). Therefore, properties of the initialization fields are still visible in the deep ocean of the forced run."

L312: "show large content of overflow water" doesn't read well.
replaced by "feature large volumes of overflow water"

L318: "without though the higher resolution is not quite sufficient" – some words or explanation is missing
adding "… and hence lacks some mixing between the boundary and interior, which is well parameterized in the non-eddying configuration."

L330: shown
corrected, thanks

L339: suffers is probably not the best word here.
replaced by "is deficient due to"

L355: Irminger Rings entering the Labrador Sea interior – maybe add some references.
There are several references about eddies in this region included in the text already (Introduction and Discussion). In this sentence we describe what we find in our simulations and thus think no additional reference is required here.

L357: are crucial
corrected

L366: dynamically active (highly not needed)
dropped "highly"

L371: What does "largely improved" really mean?
replaced by "significantly mitigate"

L375: "well seen" doesn't strike me as formal scientific wording
changed to "illustrated by"

L388: In terms of imprints of the different Northwest Corner dynamics on meltwater tracer concentrations, are there any other studies that could be referenced/included in the discussion here?
Not that we are aware of. This topic is subject to a follow-on paper we are currently working on.

L418: "the coupled-nested configurations" – plural or singular – I.e. are you meaning the control and melt experiments with this setup?
singular, only the reference state is discussed here

Figure 14 caption: States magenta lines but I see red and yellow.
the ice edge of the reference state is depicted by a magenta outline in all panels

L428: "over the entire but mostly eastern SPNA" almost feels contradictory.
true, now reads "… across the entire SPNA with a maximum on its eastern side."

L445: Other studies have looked at the role of Ekman transports around the sub-polar gyre. Would be good to reference them.
references are included in the discussion section: "As shown here but also by earlier studies \citep{SchulzeChretien2018,Castelao2019,Duyck2022} upwelling favorable Ekman transport plays a significant role in spreading relatively fresh coastal waters offshore into the Labrador and Irminger Seas."

L450: Is this realistic. Is there a concern of the atmospheric scale being too coarse to look at processes around the narrow boundary currents. This comes up later in the paper, but would be good to mention here. Also, would be good to reference those works that have previously discussed Ekman transport's role in exchange from the WGC and LC.
By intention we gathered the entire discussion on this topic in Section 4 Discussion in order to prevent redundancy. This section is really only about our results and immediate interpretation.

L474: "explored in many model studies before" – add some references to those previous studies
these are listed in the introduction, to which we now refer in the beginning of the sentence: "As we note in the introduction, …"

L476: "passed decade already" – I think the authors may mean a previous decade?
we actually meant "past" but inserted "last" now

L484: being
corrected

L487: "Potentially in consequence thereof" doesn't read well
removed; sentence was rephrased: "Recently enhanced deep convection in the Irminger Sea \citep{Ruehs2021} may have compensated a lack of deep water formation in the Labrador Sea and hence offset an impact by recently increased runoff from Greenland." (see comment by Reviewer 1)

L492: What is meant by "does neither cover"?
replaced "cover" by "include"

L494: the objective of whether...
corrected

L501: Don't like the wording "quite typical" for the salinity bias – maybe explain this in more detail and more clearly.
Removed from sentence. Since we are asked to shorten the manuscript, we prefer to not go into this discussion.

L503: remove also
removed

L505: "doubting the importance" isn't a great choice of wording
sentence rephrased: "However, we cannot exclude a significant influence by the ocean and climate mean state, which differs between coupled and forced experiments."

L509: disagree may be a better word than oppose
good suggestion, thank you, changed accordingly

L522: I think Schulze-Chretien and Frajka-Williams should also be referred to here.
yes, indeed; reference added

L527: remove also
done

L535-540: Some other studies suggest resolutions up to 1/60th degree may now be needed.
Sure, this would improve the mesoscale and even submesoscale dynamics. But it is impractical at this stage of computational power for any global, multi-decadal ocean model applications. Also, Hallberg (2013) shows that the Rossby radius in most of the Labrador Sea can be resolved by a 1/20° model.

L548: remove "a"
corrected

L548: Gillard et al, (2022) – Ocean Modelling – looks at the some impacts on this exchange when changing the vertical resolution
Thank you for pointing this out. This reference is now included: "\citet{Gillard2022} recently highlighted the importance of vertical model grid resolution and an associated improved representation of the local topography for the exchange between the WGC and the interior Labrador Sea. An aspect our nested simulations do not reflect as the vertical resolution of 46 levels is the same as in the non-eddying configuration."

L585: Not sure what "presenting with the strongest weakening" is really saying
replaced "presenting with" by "simulating"

L636: "present with diverse sensitivity" doesn't read well.
This sentence has been removed. The final statement now says: "Climate model experiments disagree on the potential impact of Greenland meltwater among all other consequences of global warming \citep[e.g.][]{Swingedouw2006,Mikolajewicz2007} but typically result in a weaker AMOC response than forced ocean models \citep{Martin2022, Swingedouw2022}. However, such coupled simulations should not be dismissed per se in favor of very high resolution ocean-only configurations as recently conveyed by \citet{Swingedouw2022}. Our results emphasize that large-scale atmosphere-ocean feedback and local winds are as important as simulating a strongly eddying ocean."

---

## Referee Report (RR1)

The authors made a good effort to improve the manuscript and I believe that the majority of my comments have been answered. The paper might still be a bit long but it will definitely contribute to improve our understanding of the fate of the increasing freshwater fluxes coming from Greenland in several model configurations. The manuscript should be accepted for publication at this stage.

---

## Author Response (AR2)

We thank the two reviewers and the editor for their positive judgement of our manuscript and appreciate the thorough reading and helpful comments provided during the review process. All of the final suggestions by reviewer #1 are included in the final version of the manuscript.

We are particularly grateful for selecting our submission as a highlight paper for Ocean Science and thank Mario Hoppema for his kind summarizing statement.